# A Closer Look at Model Collapse:
# From a Generalization-to-Memorization Perspective

**Lianghe Shi**[*]
University of Michigan
United States
`lhshi@umich.edu`

**Meng Wu**[*]
University of Michigan
United States
`wymond@umich.edu`

**Huijie Zhang**
University of Michigan
United States
`huijiezh@umich.edu`

**Zekai Zhang**
University of Michigan
United States
`zzekai@umich.edu`

**Molei Tao**
Georgia Institute of Technology
United States
`mtao@gatech.edu`

**Qing Qu**
University of Michigan
United States
`qingqu@umich.edu`

## Abstract

The widespread use of diffusion models has led to an abundance of AI-generated data, raising concerns about *model collapse*—a phenomenon in which recursive iterations of training on synthetic data lead to performance degradation. Prior work primarily characterizes this collapse via variance shrinkage or distribution shift, but these perspectives miss practical manifestations of model collapse. This paper identifies a transition from generalization to memorization during model collapse in diffusion models, where models increasingly replicate training data instead of generating novel content during iterative training on synthetic samples. This transition is directly driven by the declining entropy of the synthetic training data produced in each training cycle, which serves as a clear indicator of model degradation. Motivated by this insight, we propose an entropy-based data selection strategy to mitigate the transition from generalization to memorization and alleviate model collapse. Empirical results show that our approach significantly enhances visual quality and diversity in recursive generation, effectively preventing collapse. The source code is available at `https://github.com/shilianghe007/Model_Collapse.git`

## 1 Introduction

As generative models, such as diffusion models, become widely used for image synthesis and video generation, a large volume of generated data has emerged on the Internet. Since state-of-the-art diffusion models can generate realistic content that even humans cannot easily distinguish, the training datasets for next-generation models will inevitably contain a significant proportion of synthetic data. Figure 1 illustrates this self-consuming loop, where at each iteration[2], data generated by the current model is subsequently used to train the new model for the next iteration. Unfortunately, several recent studies have demonstrated that recursively training models on datasets contaminated by AI-generated data leads to performance degradation across these self-consuming iterations—even when synthetic data comprises only a small fraction of the dataset [1]. This phenomenon, termed *model collapse* in prior work, poses a significant threat to the future development and effectiveness of generative models.

---

[*]The first two authors contribute equally.

[2]This paper uses "iteration" to denote a full cycle of training and generation, rather than a gradient update.

39th Conference on Neural Information Processing Systems (NeurIPS 2025).

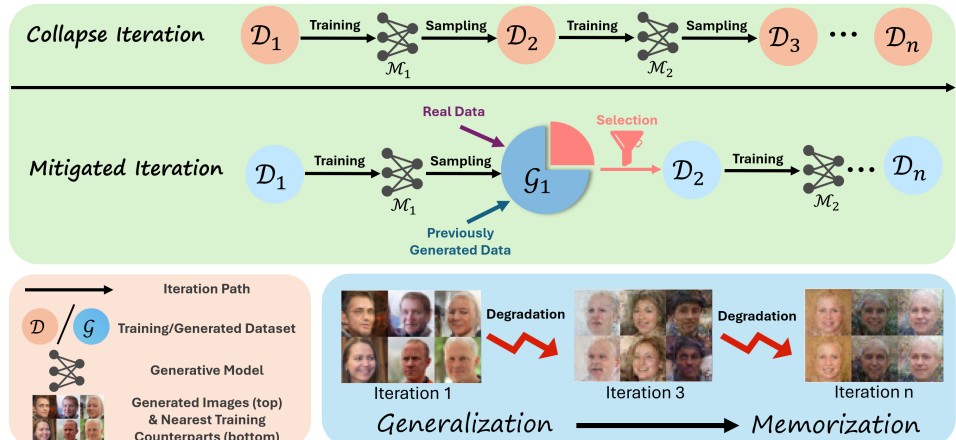

Figure 1: **High-level depiction of the self-consuming pipeline. Top:** *Collapse iteration* represents the replace paradigm where models are trained solely on synthetic images generated by the previous diffusion model. **Middle:** In the *mitigated iteration*, original real data and previously generated data are added to train the next-generation model. Our proposed selection methods construct a training subset and can further mitigate collapse. **Bottom Right:** Evolution of the generated images.

As surveyed by [2], recent studies have identified various collapse behaviors that depend on the performance metrics employed. A series of papers [3–6] reveal the model collapse phenomenon through the variance of the learned distribution. They empirically and theoretically show that the model continually loses information in the distribution tail, with variance tending towards $0$. Another line of work [5, 7–10] investigates the issue from the perspective of population risk or distribution shifts. These studies observe that the generated distribution progressively deviates from the underlying distribution, causing the model's population risk to increase throughout the recursive process. Numerous studies [11–13] also report that models begin generating hallucinated data. Despite significant theoretical insights regarding variance dynamics, the reduction of variance to negligible levels typically occurs only after an extremely large number of iterations. As noted by [2, 4], the collapse progresses at such a slow pace that it is rarely a practical concern in real-world applications. In contrast, the visual quality and diversity of generated images deteriorate rapidly. Furthermore, although population risk or distribution shifts offer a holistic view of performance degradation, they do not adequately characterize specific collapse behaviors.

Accordingly, this paper conducts an in-depth investigation into the collapse dynamics of diffusion models and identifies a *generalization-to-memorization* transition occurring across successive iterations. Specifically, during early iterations, the model demonstrates a strong capability to generate novel images distinct from those in the training set, but gradually shifts towards memorization in later iterations, merely replicating training images. This transition significantly reduces the diversity of generated content and results in higher FID scores. Moreover, directly reproducing images from training datasets may raise copyright concerns [14, 15]. Furthermore, we empirically reveal a strong linear correlation between the generalizability of the trained model and the entropy of its training dataset. As iterations progress, the entropy of the data distribution sharply decreases, directly resulting in a decline in the models generalizability, which illustrates a clear transition from generalization to memorization. Motivated by these empirical findings, we propose entropy-based selection methods to construct training subsets from candidate pools. Extensive experimental validation demonstrates that our proposed methods effectively identify high-entropy subsets, thereby decelerating the generalization-to-memorization transition. Additionally, our approach achieves superior image quality and lower FID scores in recursive training loops compared to baseline methods.

**Our Contributions.** The contributions of this paper are summarized as follows:

1. We identify the *generalization-to-memorization* transition within the self-consuming loop, providing a novel perspective for studying model collapse and highlighting critical practical issues arising from training on synthetic data.

2. We investigate the key factor driving this transition and empirically demonstrate a strong correlation between the entropy of the training dataset and the generalization capability of the model.

3. Motivated by our empirical findings, we propose entropy-based data selection methods, whose effectiveness is validated through comprehensive experiments across various image datasets.

## 2   Background

In this work, we focus on the image generation task. Let $\mathcal{X}$ be the $d$-dimensional image space, $\mathcal{X} \subseteq \mathbb{R}^d$ and let $P_0$ be a data distribution over the space $\mathcal{X}$. We use **bold** letters to denote vectors in $\mathcal{X}$. We assume the original training data $\mathcal{D}_{\text{real}} = \{\boldsymbol{x}_{\text{real}}^{(1)}, \ldots, \boldsymbol{x}_{\text{real}}^{(N_0)}\}$ are generated independently and identically distributed (i.i.d.) according to the underlying distribution $P_0$, i.e., $\boldsymbol{x}_{\text{real}}^{(i)} \sim P_0$.

**Diffusion Models.**   For a given data distribution, diffusion models do not directly learn the probability density function (pdf) of the distribution; instead, they define a forward process and a reverse process, and learn the score function utilized in the reverse process. Specifically, the forward process [16] progressively adds Gaussian noise to the image, and the conditional distribution of the noisy image is given by: $p(\boldsymbol{x_t}|\boldsymbol{x_0}) = \mathcal{N}(\boldsymbol{x_t}; \sqrt{\bar{\alpha}_t}\boldsymbol{x_0}, (1-\bar{\alpha}_t)\boldsymbol{I})$, where $\bar{\alpha}_t$ is the scale schedule, $\boldsymbol{x_0}$ is the clean image drawn from $P_0$, and $\boldsymbol{x_t}$ is the noisy image. This forward can also be described as a stochastic differential equation (SDE) [17]: $d\boldsymbol{x} = f(\boldsymbol{x},t)dt + g(t)d\boldsymbol{w}$, where $f(\cdot, t) : \mathbb{R}^d \to \mathbb{R}^d$ denotes the vector-valued drift coefficient, $g(t) \in \mathbb{R}$ is the diffusion coefficient, and $w$ is a standard Brownian motion. This SDE has a corresponding reverse SDE as $d\boldsymbol{x} = [f(\boldsymbol{x},t) - g^2(t)\nabla_{\boldsymbol{x}} \log p_t(\boldsymbol{x})]dt + g(t)d\boldsymbol{w}$, where $dt$ represents a negative infinitesimal time step, driving the process from $t = T$ to $t = 0$. The reverse SDE enables us to gradually convert a Gaussian noise to a clean image $\boldsymbol{x} \sim P_0$.

The score function $\nabla_{\boldsymbol{x}} \log p_t$ is typically unknown and needs to be estimated using a neural network $\boldsymbol{s}_\theta(\boldsymbol{x}, t)$. The training objective can be formalized as

$$\mathbb{E}_{t \sim \mathcal{U}(0,T)} \mathbb{E}_{p_t(\boldsymbol{x})} \left[ \lambda(t) \left\| \nabla_{\boldsymbol{x}} \log p_t(\boldsymbol{x}) - \boldsymbol{s}_\theta(\boldsymbol{x}, t) \right\|_2^2 \right],$$

and can be efficiently optimized with score matching methods such as denoising score matching [18].

**Self-Consuming Loop.**   Following the standard setting of model collapse [1, 4, 7, 11, 19–21], we denote the training dataset at the $n$-th iteration as $\mathcal{D}_n$. Let $\mathcal{A}(\cdot)$ denote the training algorithm that takes $\mathcal{D}_n$ as input and outputs a diffusion model characterized by the distribution $P_n$, i.e., $P_n = \mathcal{A}(\mathcal{D}_n)$. In this work, we train the diffusion model from scratch at each iteration. Subsequently, the diffusion model generates a synthetic dataset of size $N_n$, denoted by $\mathcal{G}_n \sim P_n^{N_n}$, which is used in subsequent iterations.

Based on the specific way of constructing training datasets, previous studies [4, 11, 22] distinguish two distinct iterative paradigms:

- **The replaced training dataset.** At each iteration, the training dataset consists solely of synthetic data generated by the previous diffusion model, i.e., $\mathcal{D}_n = \mathcal{G}_{n-1}$. Several studies [4, 8, 22] refer to this as the "*replace*" paradigm and have demonstrated that under this setting, the variance collapses to 0 or the population risk diverges to infinity.

- **The accumulated training dataset.** A more realistic paradigm [8, 11] is to maintain access to all previous data, thereby including both real images and all synthetic images generated thus far, i.e., $\mathcal{D}_n = (\cup_{j=1}^{n-1} \mathcal{G}_j) \cup \mathcal{D}_{\text{real}}$. However, continuously increasing the training dataset size quickly demands substantial computational resources. A practical compromise is to subsample a fixed-size subset from all candidate images, referred to as the "*accumulate-subsample*" paradigm in [4]. Under certain conditions, prior work [4, 8] have shown that accumulating real and synthetic data mitigates model degradation, preventing population risk from diverging.

This work focuses on the replace and accumulate-subsample paradigms following prior studies of [3, 4, 6]. Please refer to the Appendix for additional related work.

# 3 Model Collapses from Generalization to Memorization

In this section, we empirically demonstrate the transition from generalization to memorization that occurs over recursive iterations, and investigate the underlying factors driving this transition. All experiments in this section are conducted on the CIFAR-10 dataset [23] using a UNet-based DDPM model [16], under the *replace* paradigm, where each model is trained solely on samples generated by the model from the previous iteration. However, in Appendix D, we extend the explorations to other datasets and paradigms, where the conclusion remains valid.

**Generalization Score.** To quantify generalization ability, we adopt the *generalization score* [15, 24, 25], defined as the average distance between each generated image and its nearest training image:

$$\text{GS}(n) \triangleq \text{Dist}(\mathcal{D}_n, \mathcal{G}_n) = \frac{1}{|\mathcal{G}_n|} \sum_{\boldsymbol{x} \in \mathcal{G}_n} \min_{z \in \mathcal{D}_n} \kappa(\boldsymbol{x}, \boldsymbol{z}), \tag{1}$$

where $\kappa(\cdot, \cdot) : \mathbb{R}^d \times \mathbb{R}^d \to \mathbb{R}$ denotes a distance metric between two data points. A higher generalization score $\text{GS}(n)$ indicates that the model generates novel images rather than replicating training samples.

**Remark:** [15] measure generalizability as the probability that the similarity between a generated image and its nearest training sample exceeds a threshold. [25] assesse memorization via a hypothesis test. While definitions of generalizability vary across studies, they are fundamentally similar, relying on nearest neighbor distances.

**Highlight of Observations:** The generated data progressively collapses into numerous compact local clusters over model collapse iterations, as evidenced by both the sharp decline in entropy over iterations and visualizations. This localized concentration of data points then facilitates memorization in subsequent models, reducing their ability to generate novel images. Our claim is supported by the following three findings.

## 3.1 Finding I: Generalization-to-Memorization Transition

A generalization-to-memorization transition is revealed by experiments showing that the model initially generates novel images but gradually shifts to reproducing training samples in later iterations. We conduct iterative experiments on the CIFAR-10 benchmark [23] to illustrate this transition. Figure 2 visualizes generated samples alongside their nearest neighbors in the training set. With a relatively large sample size, i.e., 32,768 real samples as the starting training dataset, the model tends to generalize first and then memorize. At the early iterations, the diffusion model exhibits strong generalization in early iterations, producing high-quality novel images with little resemblance to training samples. However, its generalization ability deteriorates rapidly over successive iterations, and the model can only copy images from the training dataset after several iterations.

To quantitatively validate the generalization-to-memorization transition, we track the *generalization score* (GS) introduced in Equation (1), which measures the similarity between the generated images $\mathcal{G}_i$ and the corresponding training images $\mathcal{D}_i$ at each iteration. We follow the protocol of [15] and construct six nested CIFAR-10 subsets of increasing size: $|\mathcal{D}_1| \in \{1,024; 2,048; 4,096; 8,192; 16,384; 32,768\}$. These subsets span approximately $3\%$ to $65\%$ of the full training set, providing controlled start points that reflect varying degrees of memorization and generalization. As shown in Figure 2, GS drops almost exponentially with successive iterations, providing strong empirical evidence for the generalization-to-memorization transition. The decline is noticeably slower for larger training subsets, indicating that larger sample sizes preserve generalization longer and delay the onset of memorization. For the smallest dataset of $1,024$ images, the model enters the memorization regime from the first iteration and remains there throughout.

## 3.2 Finding II: The Entropy of the Training Set Shrinks Sharply over Iterations

We identify entropy as the key evolving factor in the training data that drives the transition from generalization to memorization. Prior work [24] interprets generalization in diffusion models as a failure to memorize the entire training set. [15] further show that diffusion models tend to generalize

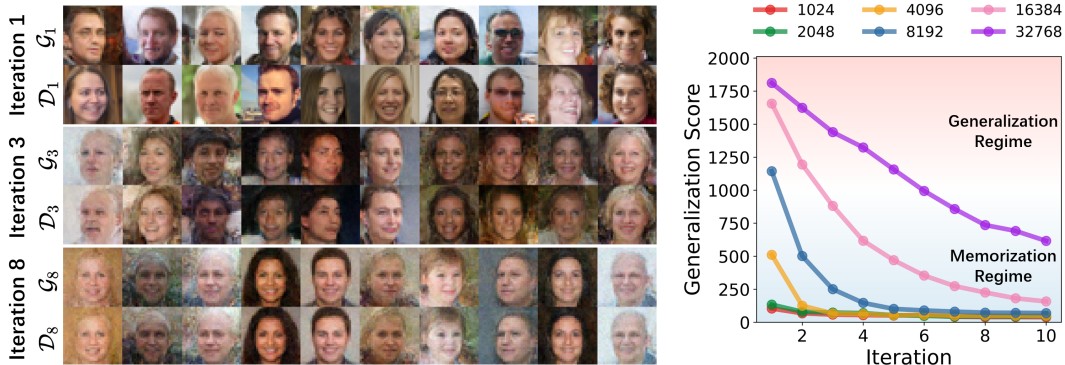

Figure 2: **The generalization-to-memorization transition. Left:** visualization of the generated images ($\mathcal{G}_n$) and their nearest neighbors in the training dataset ($\mathcal{D}_n$). As the iteration proceeds, the model can only copy images from the training dataset. **Right:** quantitative results of the generalization score of models over successive iterations. We use different colors to represent different dataset sizes. A smaller dataset has a larger decaying rate and even falls in the memorization regime at the start [15]. We use "iteration" to denote a full cycle of training and generation, rather than a gradient update.

when trained on large datasets (e.g., $> 2^{14}$ images in CIFAR-10) and to memorize when trained on small ones (e.g., $< 2^9$ images). However, since we fix the size of the training dataset for every iteration, the previous conclusion [15] that a larger dataset leads to generalization cannot fully explain the phenomenon observed in Finding I. We hypothesize that although sample size remains constant, the amount of information it contains decreases over time, making it easier for the model to memorize. Based on this hypothesis, we adopt the differential entropy to quantify the information content and complexity of the continuous image distribution.

**Definition 3.1** (Differential Entropy [26])**.** Let $\boldsymbol{X}$ be a continuous random variable with probability density function $f$ supported on the set $\mathcal{X}$. The differential entropy $H(\boldsymbol{X})$ is defined as

$$H(\boldsymbol{X}) = \mathbb{E}[-\log(f(\boldsymbol{X}))] = -\int_{\mathcal{X}} f(\boldsymbol{x}) \log f(\boldsymbol{x}) \, d\boldsymbol{x}$$

**Estimation.** However, the density function $f$ of the image distribution is unknown. We use the following Kozachenko-Leonenko (KL) estimator proposed in a well-known paper [27] to empirically estimate $H(\boldsymbol{X})$ from a finite set of i.i.d. samples $\mathcal{D}$ drawn from the distribution $P$:

$$\hat{H}_\gamma(\mathcal{D}) = \psi(|\mathcal{D}|) - \psi(\gamma) + \log c_d + \frac{d}{|\mathcal{D}|} \sum_{\boldsymbol{x} \in \mathcal{D}} \log \varepsilon_\gamma(\boldsymbol{x}), \tag{2}$$

where $\psi : \mathbb{N} \to \mathbb{R}$ is the digamma function, i.e., the derivative of the logarithm of the gamma function; $\gamma$ is any positive integer; $c_d$ denotes the volume of the unit ball in the $d$-dimensional space; and $\varepsilon_\gamma(\boldsymbol{x}) = \kappa(\boldsymbol{x}, \boldsymbol{x}_\gamma)$ represents the $\gamma$-nearest neighbor distance, where $\boldsymbol{x}_\gamma$ is the $\gamma$-th nearest neighbor of $\boldsymbol{x}$ in the set $\mathcal{D}$. Prior work [28] has shown that the KL estimator is asymptotically unbiased and consistent under broad conditions.

We use the KL estimator with $\gamma = 1$ to measure the entropy of the image dataset at each iteration. As shown in Figure 3, the entropy of the generated image dataset—used as the training set in the next iteration—consistently decreases over iterations. With a fixed dataset size, the only dataset-dependent term in Equation (2) is the sum of nearest-neighbor distances $\varepsilon(\boldsymbol{x})$ indicating that samples in $\mathcal{D}$ become increasingly concentrated. This suggests the distribution is becoming spiky. Figure 3 further illustrates this trend by projecting high-dimensional images onto the subspace spanned by their top two eigenvectors. The visualization reveals that the generated images form numerous local clusters, while the overall support of the distribution remains relatively stable.

### 3.3 Finding III: The Correlation between Entropy and Generalization Score

We verify that the generalization score of the trained model is strongly correlated with the entropy of the training dataset. The similar collapsing trends of entropy and the generalization score motivate a

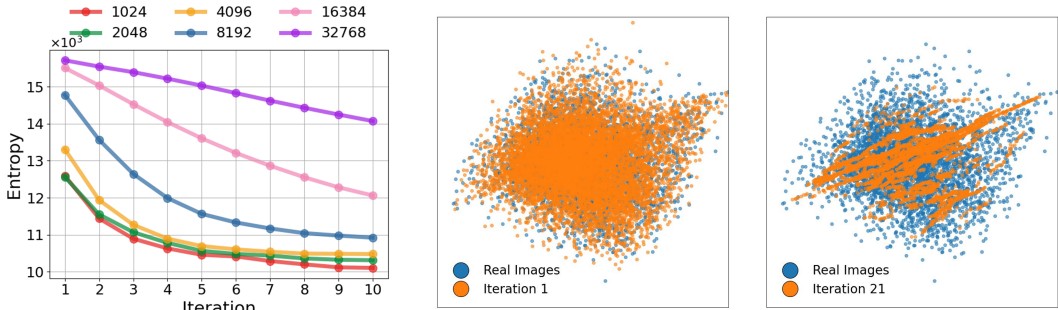

Figure 3: **Decreasing entropy and visualizations. Left:** The evolving entropy of the training dataset over iterations. Under the replace paradigm, the training data is generated data from the last iteration. **Middle** and **Right:** 2-D projection of data points onto the first two singular bases of the real dataset. The orange points represent the generated images at the 1-st and 21-st iterations, respectively.

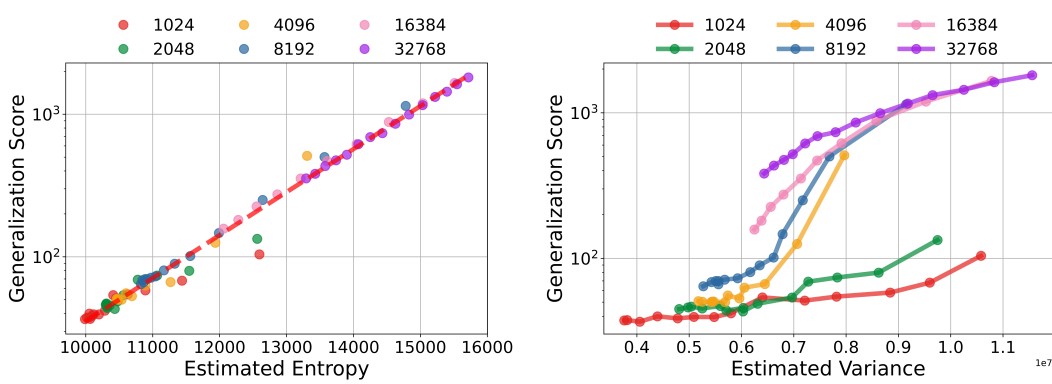

(a) Generalization score versus estimated entropy.  (b) Generalization score versus trace of covariance.

Figure 4: **Scatter plots** of the generalization score and properties of the training dataset, i.e., entropy and variance. Each point denotes one iteration of training in the self-consuming loop. We use different colors to represent the results of different dataset sizes.

deeper investigation into their relationship. In Figure 4a, we present a scatter plot of entropy versus generalization score across different dataset sizes and successive iterations, with the y-axis shown on a logarithmic scale. Notably, the entropy of the training dataset exhibits a significant linear relationship with the logarithm of the generalization score. The Pearson correlation coefficient is $0.91$ with a $p$-value near zero, quantitatively confirming the strength of this correlation. Furthermore, the scatter points corresponding to different dataset sizes are all approximately aligned along a single line, suggesting the generality of the relationship between entropy and generalization. Specifically, training datasets with higher entropy consistently yield better generalization in the trained model. We also observe that larger datasets typically have higher entropy and result in better generalization, aligning with the conclusion in [15] regarding the connection between dataset size and generalization. For comparison, Figure 4b shows the scatter plot of variance versus generalization score. The correlation appears substantially weaker than in Figure 4a, suggesting that variance in the training dataset may not directly influence the generalization performance of the trained model.

**Conclusion.** Findings IIII collectively indicate that the generated data gradually collapses into compact clusters, as shown by declining entropy. This concentration promotes memorization in the model of the next iteration and reduces its ability to generalize. In Appendix D, we further extend these explorations to the FFHQ dataset and accumulate paradigm, demonstrating a more robust relation.

# 4 Mitigating Model Collapse with Data Selection Methods

In this section, we propose a sample selection method that selects a subset of images from the candidate pool to mitigate the model collapse. Motivated by the empirical finding in Section 3, the selected training images should have high entropy, characterized by large nearest-neighbor distances and greater diversity. This objective can be formalized as the following optimization problem:

$$\max_{\mathcal{D} \subset \mathcal{S}, \, |\mathcal{D}|=N} \hat{H}_1(\mathcal{D}) \quad \Longleftrightarrow \quad \max_{\mathcal{D} \subset \mathcal{S}, \, |\mathcal{D}|=N} \sum_{\boldsymbol{x} \in \mathcal{D}} \log \min_{\boldsymbol{y} \in \mathcal{D} \setminus \boldsymbol{x}} \kappa(\boldsymbol{x}, \boldsymbol{y}),$$

where $\mathcal{S}$ denotes the candidate pool. For the accumulate-subsample setting, $\mathcal{S}$ is all the accumulated images so far. In the replace setting, $\mathcal{S}$ consists of the images generated by the previous model, with size twice that of the target training set.

Unfortunately, this non-convex max-min problem is hard to solve to a global solution efficiently. Alternatively, we approximate the solution through a greedy strategy inspired by farthest-point sampling, which aims to maximize the nearest-neighbor distance.

**Method I: Greedy Selection.** The procedure iteratively constructs a subset $\mathcal{D} \subset \mathcal{S}$ of size $n$ by adding the farthest point one at a time as follows:

1. **Initialization:** Randomly select an initial point from the dataset $\mathcal{S}$ and add it to the set $\mathcal{D}$.
2. **Iterative Selection:** At each iteration, for every candidate point $\boldsymbol{x} \in \mathcal{S} \setminus \mathcal{D}$, compute the minimum distance from $\boldsymbol{x}$ to all points currently in $\mathcal{D}$. Select the point with the *maximum* of these minimum distances and add it to $\mathcal{D}$, i.e., $\boldsymbol{x}_{select} = \arg\max_{\boldsymbol{x} \in S \setminus \mathcal{D}} \min_{\boldsymbol{y} \in \mathcal{D}} \kappa(\boldsymbol{x}, \boldsymbol{y})$.
3. **Termination:** Repeat the selection process until $|\mathcal{D}| = N$.

The Greedy Selection method can efficiently and effectively extract a subset with a large entropy. However, this greedy method carries a risk of over-optimization, which may lead to an excessively expanded distribution and a progressively increasing variance in the selected samples. To mitigate this, we also provide the following Threshold Decay Filter, which can control the filtration strength by a decaying threshold.

**Methods II: Threshold Decay Filter.** The procedure constructs a subset $\mathcal{D} \subset \mathcal{S}$ of size $N$ by iteratively selecting samples that are sufficiently distant from the current set $\mathcal{D}$. The algorithm proceeds as follows:

1. **Initialization:** Set an initial threshold $\tau > 0$. Randomly select one point from the dataset $\mathcal{S}$ and add it to the set $\mathcal{D}$.
2. **Threshold-based Selection:** For each point $\boldsymbol{x} \in \mathcal{S} \setminus \mathcal{D}$, compute the distance from $\boldsymbol{x}$ to all points in $\mathcal{D}$. If all distances are greater than the current threshold $\tau$, add $\boldsymbol{x}$ to $\mathcal{D}$.
3. **Threshold Decay:** If $|\mathcal{D}| < N$ after a complete pass through $\mathcal{S} \setminus \mathcal{D}$, reduce the threshold $\tau$ by multiplying it with a decay factor $\alpha \in (0, 1)$, and repeat Step 2.
4. **Termination:** Repeat Steps 2–3 until $|\mathcal{D}| = N$.

Threshold Decay Filter is a soft variant of Greedy Selection that provides adjustable control over the selection strength. When the initial threshold is set sufficiently high and the decay factor is close to 1, the Threshold Decay Filter behaves similarly to Greedy Selection. Conversely, if both the initial threshold and decay factor are set to 0, the filter does not filter out any data point and reduces to the vanilla replace or accumulate-subsample paradigm. In practice, we first extract image features using a DINOv2 [29] model and compute distances in the feature space, i.e., $\kappa(\boldsymbol{x}, \boldsymbol{y}) = \|h(\boldsymbol{x}) - h(\boldsymbol{y})\|_2$, where $h(\cdot)$ denotes the feature extractor.

# 5 Experiments

We empirically evaluate how our data selection strategies introduced in Section 4 interact with the two self-consuming paradigms: *replace* and *accumulate-subsample*. In both settings, our method serves as a plug-in component for selecting high-quality and diverse training samples from the candidate pool. We demonstrate that the proposed strategies effectively alleviate memorization and

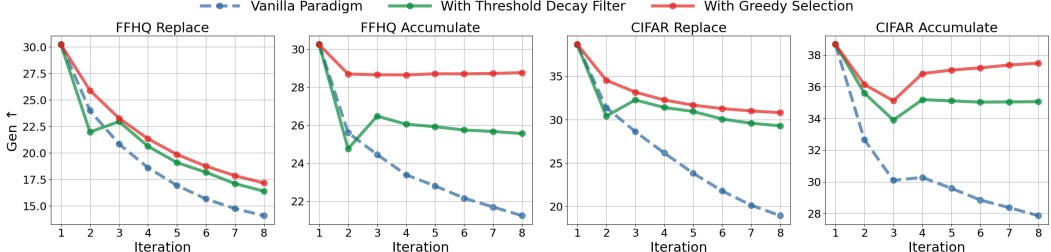

Figure 5: **Generalization Score of the trained model over iterations.** We indicate the settings on top of the subfigures. In each subfigure, three different lines are used to represent the vanilla paradigm and its variants augmented with the proposed selection methods.

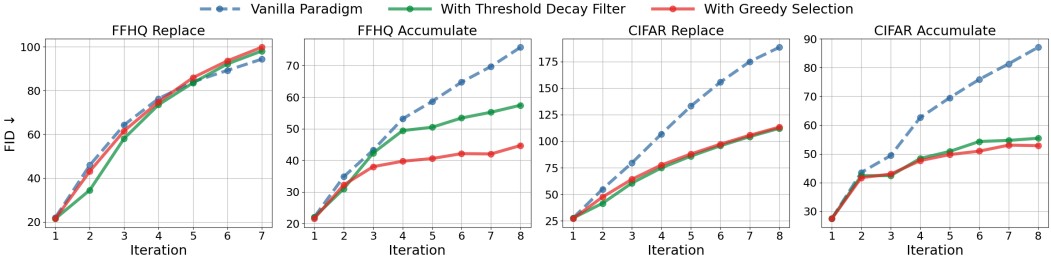

Figure 6: **FID of the generated images over iterations.** We indicate the settings on top of the subfigures. In each subfigure, three different lines are used to represent the vanilla paradigm and its variants augmented with the proposed selection methods.

reduce FID scores, thereby mitigating model collapse. Additionally, experiments on classifier-free guidance (CFG) [30] generation show that our method effectively mitigates diversity collapse of CFG.

**Datasets.** We conduct experiments on three widely used image generation benchmarks. **CIFAR-10** [23] consists of $32 \times 32$ color images in $10$ classes. Due to computational constraints, we use a subset of 32,768 training images. Our goal is not to achieve state-of-the-art FID among large diffusion models but rather to demonstrate that our method mitigates memorization in the self-consuming loop. As shown in Section 3, this subset is sufficient to observe the transition from generalization to memorization. We also conduct experiments on subsets of **FFHQ** [31], downsampled to $32 \times 32$ resolution, and **MNIST** [32], using 8,192 and 12,000 training images, respectively.

**Model.** For CIFAR-10 and FFHQ, we employ a UNet-based backbone [33] designed for $32 \times 32$ RGB images, which predicts noise residuals. The network contains approximately 16M parameters. For MNIST, we use a similar UNet-based architecture adapted for single-channel inputs, with a total of 19M parameters. Detailed network configurations are provided in the Appendix.

**Implementation.** For iterative training and sampling, our implementation is based on the Hugging Face Diffusers codebase [34] of DDPM. We use a mixed-precision training of FP16 to train the models. We adopt an Adam optimizer with a learning rate of $10^{-4}$ and a weight decay of $10^{-6}$. The batch size is $128$. A 1000-step denoising process is used, with all other hyperparameters set to their default values. For Threshold Decay Filter, we use an initial threshold of $60$ and a decay rate of $0.95$. We show in the Appendix that our method is robust in a wide range of hyperparameters.

**Evaluation Metrics.** We use the generalization score and entropy to evaluate the effectiveness of our method in mitigating memorization. Additionally, we adopt the Fréchet Inception Distance (FID) [35] as a metric to quantify the distributional divergence between generated images and real images.

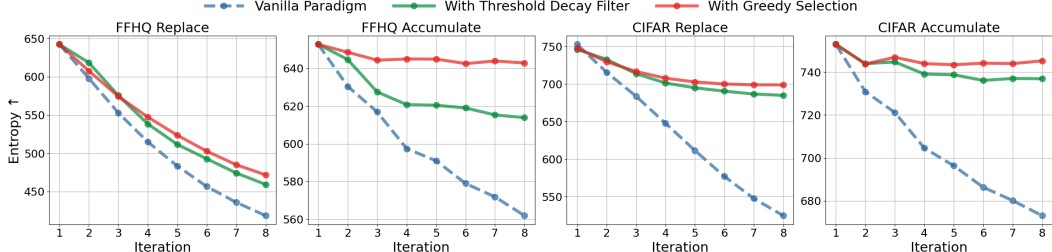

Figure 7: **Estimated entropy of the training datasets over iterations.** We indicate the settings on top of the subfigures. In each subfigure, three different lines are used to represent the vanilla paradigm and its variants augmented with our selection methods.

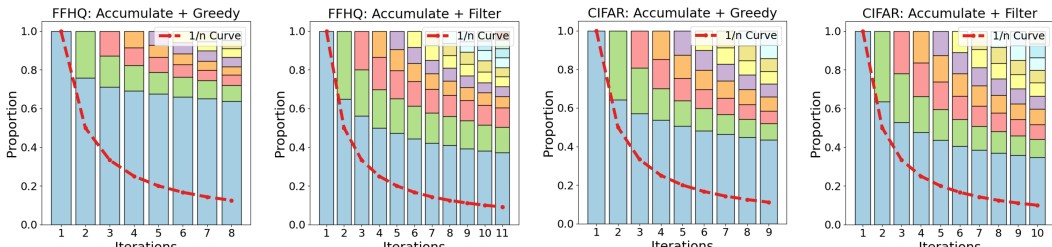

Figure 8: **Proportion of the selected images** from previous iterations or the real dataset. We use different colors to represent different sources. Particularly, the blue bars denote the proportion of the real images. The red line represents the $1/n$ curve that indicates the proportion of the real images if we evenly select the data subset from all available images (accumulate-subsample). We indicate the settings on top of the subfigures.

**Results.** To evaluate the efficacy of our selection methods in mitigating memorization within the self-consuming loop, we compare the generalization scores of two vanilla paradigms with those augmented by Greedy Selection or Threshold Decay Filter. As shown in Figure 5, Greedy Selection consistently improves generalization scores across all datasets and paradigms and is particularly effective in the accumulate-subsample setting. Importantly, our selection methods mitigate memorization without compromising FID performance; in fact, they can even slow down FID degradation. Figure 6 reports the FID of generated images over successive iterations. Under the accumulate-subsample paradigm, both methods yield notable improvements in FID, with Greedy Selection outperforming Threshold Decay Filter. For example, the vanilla accumulate paradigm reaches an FID of 75.7 at iteration 8, whereas Greedy Selection significantly reduces it to 44.7. On the FFHQ dataset under the replace paradigm, however, Threshold Decay Filter performs better, suggesting that adaptive selection strength may be beneficial in certain cases.

**Analysis for the Improvement.** We present the estimated entropy of the training datasets in Figure 7. As shown in the figure, our selection methods effectively increase the entropy of the training data at each iteration, consistent with their design. These more diverse, higher-entropy datasets subsequently enable the next-iteration model to generalize better, as evidenced by the improved generalization scores in Figure 5.

We further investigate which samples are selected by our methods under the accumulate-subsample paradigm, where the model has access to all prior synthetic images and the real images, but is trained on a subset of them. Figure 8 shows the proportion of selected samples originating from different sources, with the blue bar indicating the proportion of real images. As illustrated in the figure, both Greedy Selection and Threshold Decay Filter consistently select a significantly higher proportion of real images compared to the $1/n$ reference curve. For example, on the FFHQ dataset, Greedy Selection selects 65% real images at the 8-th iteration, while vanilla subsampling includes only 12.5%. This outcome arises because the image distribution progressively collapses into compact clusters over iterations, and our selection methods tend to prioritize boundary samples—namely, real images—by maximizing the nearest neighbor distance.

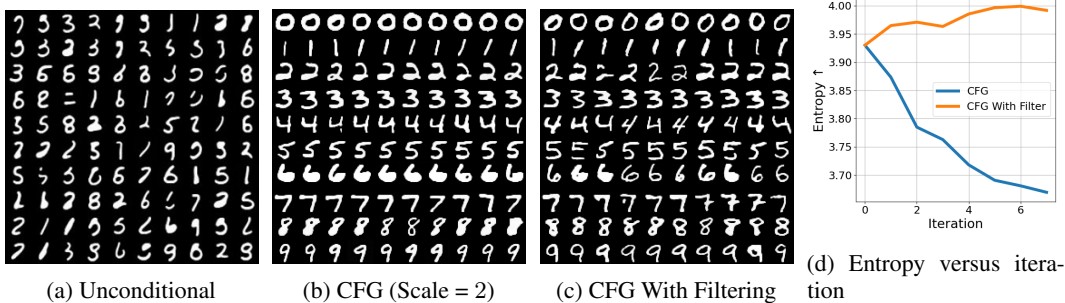

| (a) Unconditional | (b) CFG (Scale = 2) | (c) CFG With Filtering | (d) Entropy versus iteration |

Figure 9: **Comparisons of unconditional, CFG, and CFG augmented with filter method.** (a)-(c): Generated samples on MNIST [32] at the 8-th iteration under the accumulate paradigm. The **FIDs** of images in (a)-(c) are 74.4, 66.2, and **22.4** respectively. (d): The estimated entropy of the generated dataset over iterations, which reflects the diversity of the generated images.

**Diversity Improvement on Classifier-Free Diffusion Guidance (CFG) [30].** We further validate the effectiveness of our methods in the CFG setting and show that they substantially improve the diversity of generated images. CFG is a widely used conditional generation technique that consistently enhances perceptual quality but often sacrifices diversity [30, 36, 37]. Prior work by [38] identifies the CFG scale as a key factor influencing the rate of model collapse and suggests that setting a moderate scale can help mitigate model collapse. Experimentally, we observe that the CFG generation can indeed generate clearer images than the unconditional baseline, as compared in Figures 9a and 9b. However, the diversity of generated images collapses rapidly even with a modest guidance scale, with samples within each class soon becoming nearly identical. In contrast, Figures 9c and 9d demonstrate that augmenting CFG with our data selection methods significantly improves image diversity and yields substantially lower FID scores compared to the vanilla CFG paradigm. These results demonstrate that our approach effectively mitigates the diversity loss of CFG while preserving its quality advantage throughout the self-consuming loop.

## 6 Conclusion and Discussion

In this work, we reveal a generalization-to-memorization transition in diffusion models under recursive training, highlighting a serious practical concern and offering a new perspective on model collapse. We empirically demonstrate that the entropy of the training data decreases over iterations and is strongly correlated with the model's generalizability. Motivated by the findings, we propose an entropy-based data selection strategy that effectively alleviates the generalization-to-memorization transition and improves image quality, thus mitigating model collapse.

Based on this work, we believe many future directions could be further investigated. This paper doesn't rigorously explain why the entropy is collapsing. The finite dataset size, training and sampling errors, and the model bias could all contribute to the collapsing entropy (or the information loss). We envision that a more formal theoretical analysis of collapse dynamics could be developed based on theoretical models such as the mixture of Gaussians [39–41]. Besides, this framework could be extended to the language modality, investigating the discrete diffusion model [42–44]. Additionally, a more efficient algorithm is needed, as the current greedy selection method is computationally expensive.

## Acknowledgement

LS, MW, HZ, ZZ, and QQ acknowledge funding support from NSF CCF-2212066, NSF CCF-2212326, NSF IIS 2402950, and ONR N000142512339, and the Google Research Scholar Award. MT is grateful for partial supports by NSF Grants DMS-1847802, DMS-2513699, DOE Grants NA0004261, SC0026274, and Richard Duke Fellowship. We also thank Prof. Peng Wang (University of Michigan and University of Macau), Mr. Xiang Li (University of Michigan), and Mr. Xiao Li (University of Michigan) for fruitful discussions and valuable feedback.

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

# A  Related Work

## A.1  Model Collapse

As state-of-the-art generative models continue to improve in image quality, AI-generated images have become increasingly indistinguishable from real ones and are inevitably incorporated into the training datasets of next-generation models. In fact, [11] have shown that the LAION-5B dataset [45], which is used to train Stable Diffusion, indeed contains synthetic data. Unfortunately, recent studies [1, 4, 7, 11, 19–21] demonstrate that model performance deteriorates under such recursive training, potentially leading to model collapse, where the model generates increasingly homogeneous and meaningless content. The concept of model collapse was first introduced in [3], which also provides a theoretical framework based on Gaussian models. Their analysis shows that if a Gaussian model is recursively estimated using data generated by its predecessor, its variance converges to zero—collapsing the distribution into a delta function concentrated at a single point.

Following this important line of work, substantial research has further explored the **model collapse phenomenon**. [6] extend the theoretical analysis from Gaussian to Bernoulli and Poisson models. [7] study recursive training under high-dimensional linear and ridge regression settings, providing a linear error rate and proposing optimal regularization to mitigate collapse. [20] argue that the conventional scaling laws for foundation models break down when synthetic data is incorporated into the training set. [46] demonstrate that model biases are amplified through recursive generation and proposes algorithmic reparation techniques to eliminate such biases and negative semantic shifts. [47] introduce a statistical model that characterizes the collapse process in language models and theoretically estimates the maximum allowable proportion of synthetic data before collapse occurs, validated empirically. [48] show that human preferences can be amplified in an iterative loop: modeling human curation as a reward process, the curated distribution $p_t$ converges to $p^*$ that maximizes the expected reward as $t \to \infty$. In the work of [48], the reward $r(x)$ is a pointwise function over individual images, whereas our method can be viewed as a curation strategy that maximizes dataset-level entropy at each iteration. Whether the theory in [48] extends to dataset-wise rewards $r(S)$, such as dataset entropy, remains an open question. Nonetheless, their intuition aligns with our results that entropy can be iteratively enhanced compared to vanilla methods. [49] provide the first generalization error bounds for Self-Consuming Training Loops (STLs), showing that mitigating model collapse requires maintaining a non-negligible portion of real data and ensuring recursive stability. In contrast, our work focuses on a specific collapse phenomenon—the generalization-to-memorization transition—and introduces an entropy-based data selection algorithm to mitigate this behavior. [50] provide a rigorous theoretical analysis for the model collapse in rectified flow models and then propose methods to mitigate it. Finally, [51] investigate collapse in diffusion models by analyzing the training dynamics of a two-layer autoencoder optimized via stochastic gradient descent, showing how network architecture shapes the model collapse.

**To mitigate model collapse**, several strategies have been proposed in prior work. One common approach is to accumulate all previously generated samples along with real data into the training set—referred to as the accumulate paradigm in our paper. This strategy has been both empirically and theoretically validated. For example, [8] show that the accumulate paradigm prevents divergence of test error under a linear regression setup. [22] further establish the universality of the error upper bound across a broad class of canonical statistical models. In a related setting, [4] study an accumulate-subsample variant, confirms that the test error plateaus and examines interactions between real and synthetic data. [5] provide theoretical guarantees for stability in iterative training, assuming the initial model trained on real data is sufficiently accurate and the clean data proportion in each iteration remains high. Despite these findings, recent work [9] present a robust negative result, showing that model collapse persists even when real and synthetic data are mixed. Similarly, [11] argue that the accumulate paradigm merely delays, rather than prevents, collapse. Introducing fresh real data in each iteration may be necessary for long-term robustness. Other complementary approaches include verification [21] and re-editing [52]. For instance, [21] theoretically underscore the importance of data selection, though it does not propose a specific method. [52] target language models and introduces a token-level editing mechanism with theoretical guarantees. Compared to prior work, this paper proposes a novel entropy-based data selection method for diffusion models that improves both generalizability and image quality, thereby mitigating model collapse.

Over the past few years, extensive research has explored **model collapse from various perspectives and dimensions**. An important study [2] makes a thorough survey about different definitions and

patterns investigated in previous work. A prominent line of work [3–6] focuses on the phenomenon of variance collapse, both empirically and theoretically demonstrating that models progressively lose information in the distributions tail, with variance tending toward zero. Another series of studies [5, 7–10] investigates model collapse through the lens of population risk and distributional shift, observing that the generated distribution increasingly diverges from the true data distribution, leading to rising population risk across recursive training cycles. Moreover, several works [11–13] report that models begin to generate hallucinated or unrealistic data as collapse progresses. [20] further suggest that the inclusion of synthetic data alters the scaling laws of model performance. In addition, [51] study the progressive mode collapse [53] in diffusion models, where the number of modes in the generated distribution gradually decreases. In this paper, we introduce a novel perspective for analyzing model collapse in diffusion models by uncovering a generalization-to-memorization transition. We show that this transition is closely tied to the entropy of the training dataset, which serves as a key indicator of model generalizability. Our findings further motivate the development of entropy-based data selection strategies that effectively mitigate model collapse.

## A.2    Generalization and Memorization

Recent studies [15, 24] have identified two distinct learning regimes in diffusion models, depending on the size of the training dataset and the model's capacity: (1) Memorization regime, when models with sufficient capacity are trained on limited datasets, they tend to memorize the training data; and (2) Generalization regime, as the number of training samples increases, the model begins to approximate the underlying data distribution and generate novel samples. To investigate the transition between these regimes, [54] show that the number of training samples required for the transition from memorization to generalization scales linearly with the intrinsic dimension of the dataset. In addition, the analysis of training and generation accuracies in [55] provides a potential step toward quantifying generalization. [56] propose a theoretically grounded and computationally efficient metric, Probability Flow Distance (PFD), to measure the generalization ability of diffusion models. Specifically, PFD quantifies the distance between distributions by comparing their noise-to-data mappings induced by the probability flow ODE. Meanwhile, concurrent work also explores memorization and generalization separately. To understand generalization, studies such as [57, 58] attribute the generalization to implicit bias introduced by network architectures. Other works study the generalized distribution using Gaussian models [59, 60] and patch-wise optimal score functions [61, 62]. As for memorization, it is investigated in both unconditional and conditional [63], as well as the text-to-image diffusion models [64, 65]. Additionally, methods to mitigate memorization in diffusion models have been proposed in [66, 67]. Distinct from prior work, our study is the first to establish a connection between model collapse and the transition from generalization to memorization. This connection not only offers a novel perspective to understand model collapse but also provides insights to mitigate it by mitigating memorization.

## A.3    Data Selection

[68–70] focus on data pruning techniques, but not necessarily in the context of self-consuming loops. While both their approaches and ours demonstrate the benefits of data selection, there are several key differences:

- Objective and evaluation: [68–70] primarily study how pruning improves scaling laws, achieving higher accuracy as dataset size varies. In contrast, our work examines how model performance evolves over an iterative training loop with a fixed dataset size, focusing on the generalization-to-memorization transition.

- Task and criteria: Prior works focus on classification tasks, where sample selection depends on label information. Our method targets generative modeling, where selection is based on dataset entropy rather than label-driven criteria. Our objectives also differ: prior work emphasizes classification accuracy, while we address model collapse from a generalization perspective, providing a different angle of analysis.

- Pruning strategy: Methods in [68–70] largely rely on per-sample evaluation, whereas our approach considers global dataset structure and relationships between samples. While this may lead to increased computational complexity, it opens up new possibilities for designing pruning criteria beyond per-sample evaluation.

- Entropy definition: In [70], while the authors use a generative model to sample images, the goal is to improve the classification performance of a classifier. And the entropy used in their method is defined in the prediction space of a classifier, which is different from the entropy of a dataset measured in our work.

# B    Experimental Details

## B.1    Network Structure

For CIFAR-10 and FFHQ, we use a UNet-based backbone, taking RGB images as inputs and predicting noise residuals. Our implementation is based on the Hugging Face Diffusers base code [34]. The architecture hyperparameters of the neural network are listed as follows:

- The numbers of in-channel and out-channel are 3.
- The number of groups for group normalization within Transformer blocks is 16.
- The number of layers per block is 2.
- The network contains 6 down-sampling blocks and 6 up-sampling blocks.
- The numbers of feature channels for the 6 blocks are $48, 48, 96, 96, 144, 144$ respectively.

For MNIST, we use a similar UNet-based backbone, taking single-channel images as inputs and predicting noise residuals. The architecture hyperparameters of the network are listed as follows:

- The numbers of in-channel and out-channel are 1.
- The number of groups for group normalization within Transformer blocks is 32.
- The number of layers per block is 2.
- The network contains 4 down-sampling blocks and 4 up-sampling blocks.
- The numbers of feature channels for the 4 blocks are $64, 128, 256, 512$ respectively.

## B.2    Implementation Details

In this paper, we use DDPM as our generative method. For efficiency, we use the FP-16 mixing precision to train our models, which is inherently implemented by the Hugging Face Diffusers codebase. The batch size of training all datasets is set to be 128. A 1000-step denoising process is used as the reverse process. For CIFAR-10, the epoch number is set to be 500; for FFHQ, the epoch number is set to be 1000. We use the Adam optimizer with a learning rate of $10^{-4}$ and a weight decay of $10^{-6}$. Other experimental hyperparameters are exactly the default values in the original Hugging Face Diffusers codebase. We use the DINOv2 model [29] to extract features of images and then calculate the distance between two images in the feature space. We use the InceptionV3 model [71] to extract features of images to calculate the FID score. All experiments are conducted on a single NVIDIA A-100 GPU.

## B.3    Pseudo-codes for the Algorithms

We present the pseudo-code of the Greedy Selection and Threshold Decay Filter methods in Algorithms 1 and 2.

# C    Ablation Study

## C.1    Training on More Samples

In Section 5 of the main paper, we augment the vanilla replace paradigm with our data selection methods. Specifically, under the replace setting, the vanilla baseline generates $N$ images and trains the next-iteration model based solely on these $N$ images from the previous iteration. Instead, the data selection methods generates $2N$ images at each iteration, and select a subset of $N$ images from the $2N$ images for training the next-iteration model. One nature question is: what if we also generate $2N$ images and then train the next-iteration model on the entire $2N$-image dataset without filtering.

---

**Algorithm 1** Greedy Selection

---

**Require:** Dataset $\mathcal{S}$, target size $N$, distance function $\kappa(\cdot, \cdot)$
**Ensure:** Selected subset $\mathcal{D}$ of size $N$
 1: Initialize $\mathcal{D} \leftarrow \{\text{random point from } \mathcal{S}\}$
 2: **while** $|\mathcal{D}| < N$ **do**
 3:     **for all** $x \in \mathcal{S} \setminus \mathcal{D}$ **do**
 4:         Compute $d(x) \leftarrow \min_{y \in \mathcal{D}} \kappa(x, y)$
 5:     **end for**
 6:     $x_{\text{select}} \leftarrow \arg\max_{x \in \mathcal{S} \setminus \mathcal{D}} d(x)$
 7:     $\mathcal{D} \leftarrow \mathcal{D} \cup \{x_{\text{select}}\}$
 8: **end while**
 9: **return** $\mathcal{D}$

---

---

**Algorithm 2** Threshold Decay Filter

---

**Require:** Dataset $\mathcal{S}$, target size $N$, initial threshold $\tau > 0$, decay factor $\alpha \in (0, 1)$, distance function $\kappa(\cdot, \cdot)$
**Ensure:** Selected subset $\mathcal{D}$ of size $N$
 1: Initialize $\mathcal{D} \leftarrow \{\text{random point from } \mathcal{S}\}$
 2: **while** $|\mathcal{D}| < N$ **do**
 3:     added $\leftarrow$ **false**
 4:     **for all** $x \in \mathcal{S} \setminus \mathcal{D}$ **do**
 5:         Compute $d_{\min}(x) \leftarrow \min_{y \in \mathcal{D}} \kappa(x, y)$
 6:         **if** $d_{\min}(x) > \tau$ **then**
 7:             $\mathcal{D} \leftarrow \mathcal{D} \cup \{x\}$
 8:             added $\leftarrow$ **true**
 9:         **end if**
10:         **if** $|\mathcal{D}| = N$ **then**
11:             **return** $\mathcal{D}$
12:         **end if**
13:     **end for**
14:     **if** not added **then**
15:         $\tau \leftarrow \alpha \cdot \tau$                                   ▷ No point added $\Rightarrow$ decay threshold
16:     **end if**
17: **end while**
18: **return** $\mathcal{D}$

---

Next, we show that when training on the whole $2N$ dataset, the performance (FID score) of the model is between the vanilla replace setting (generating $N$ images and training on those $N$ images) and the data selection method (generating $2N$ images and training on the selected $N$-images subset). The results on CIFAR-10 are shown in Figure 10.

Several conclusions can be drawn by comparing the results across these settings.

- Incorporating more data into the training-sampling recursion can indeed mitigate the rate of model collapse. Compared to the vanilla replacement paradigm (i.e., the first line), using $2N$ images (second line) yields improved performance. This aligns with prior findings [4, 10] that sample size is a key factor influencing the collapse rate.

- Further augmenting the training data with our selection method leads to even better performance than training on the full set of $2N$ images. The results validate the effectiveness of our selection method, achieving a better FID performance while largely decreasing the training budget.

In fact, there is a trade-off for the filter ratio. There are two clear extremes:

- If the ratio is 1, all 2N generated images are used for training. As we show in Figure 10, it still degrades faster than filtering (ratio equals to $1/2$)

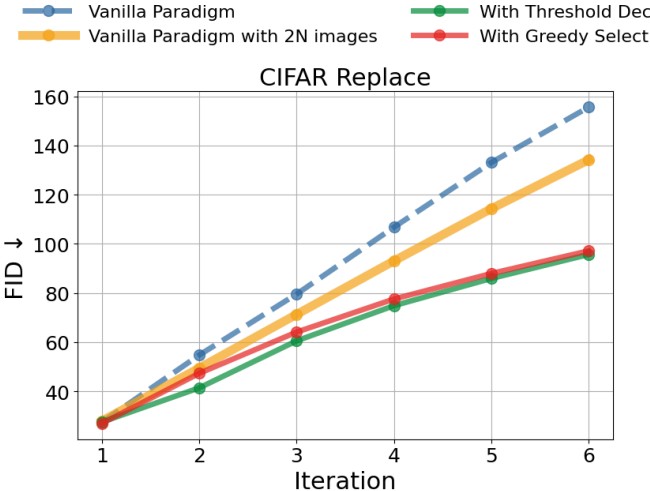

Figure 10: **FID score of the generated images over iterations.** We add an additional setting of generating $2N$ images and training the next model on the entire $2N$-images data without filtering.

| Ratio | 0.2 | 0.4 | 0.6 | 0.8 | 1.0 |
|-------|-----|-----|-----|-----|-----|
| FID | 56.0 | 44.7 | 40.0 | 38.2 | 49.5 |

Table 1: FID comparison of the models. We use various filter ratios to get the training subsets for those models.

- Conversely, if the ratio approaches 0, too few images are selected for training, which detrimentally starves the model of data.

We then use different filter ratios to get training subsets from the 2N generated images and then train models on those filtered datasets. The results in Table 1 show that an intermediate ratio yields the best FID performance.

## C.2 Different Decay Rates

The decay rate is one important hyperparameter for the Threshold Decay Filter. In this section, we use different decay rates and show that the filter is robust to a wide range of decay rates.

Figure 11 shows the FID scores of generated CIFAR-10 images across iterations for different methods. As shown in the figure, the data selection methods consistently outperform the vanilla accumulate paradigm, indicating strong robustness of the hyperparameter.

## D Additional Results

### D.1 The Generalization-to-Memorization Collapse on Accumulate Paradigm

This subsection presents additional results on the accumulate paradigm and FFHQ dataset as a supplement to the explorations in Section 3. Specifically, we show in Figure 12 that the generalization-to-memorization collapse also occurs on the accumulation paradigm. We note that model collapse has different definitions in previous studies. Here, we clarify that our results show the generalization score consistently decreases over early iterations. However, we cannot determine whether the score eventually collapses to zero or converges to a lower bound in the accumulate paradigm, as we only have fewer than 10 iterations—insufficient to draw conclusions about convergence.

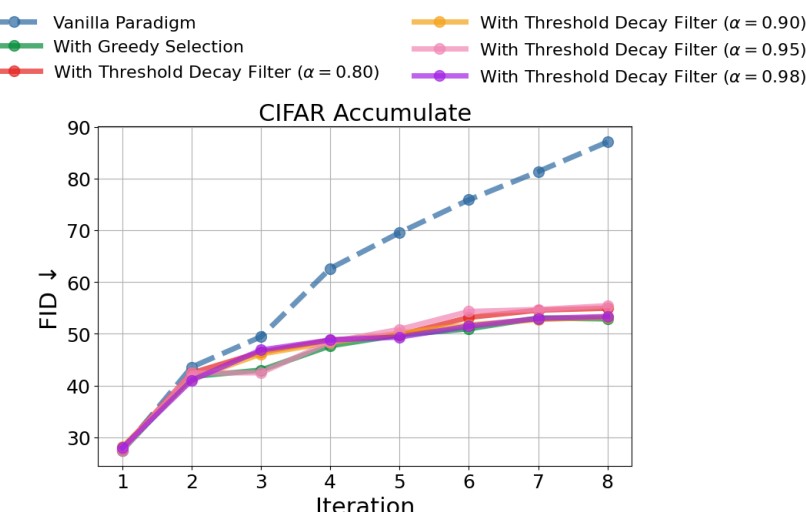

Figure 11: **FID score of the generated images over iterations.** We compare the results for different decay rates on CIFAR-10 dataset.

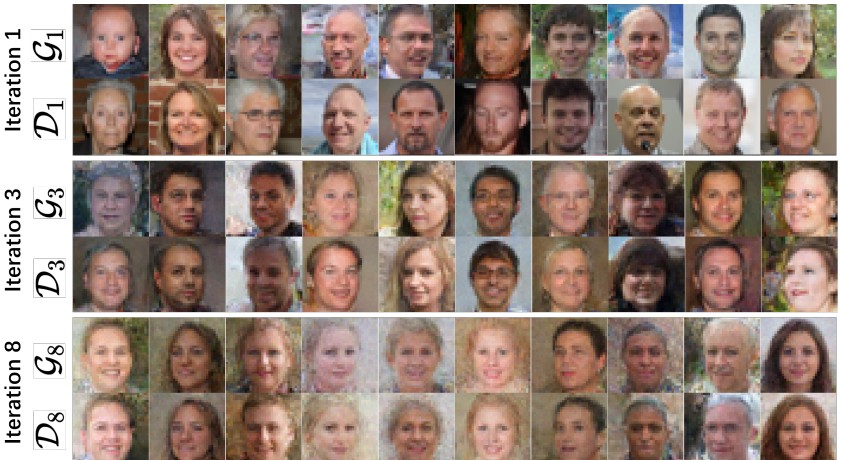

Figure 12: **The generalization-to-memorization transition on the accumulate paradigm.** Similar to Figure 2, we visualize the generated images and their nearest neighbors in the training dataset. As the iteration proceeds, the model loses generalization ability, collapsing into the memorization regime.

## D.2 The Robust Relationship between Generalization Score and Entropy

In Figure 2, we plot the results on CIFAR-10 datasets, demonstrating the dataset size-independent relationship between generalization score and estimated entropy. This subsection further incorporates the results from FFHQ and the accumulate paradigm to show a more general relation.

Specifically, we conduct the self-consuming loop for FFHQ with dataset sizes of $8,192$, $16,384$, and $32,768$. Besides, we also include the results from Appendix D.1. Combining those, we show a more robust relationship in Figure 13, where all the points align around the red dashed line. This result suggests that the entropy of the training dataset is a dataset-independent indicator for the generalization score.

## D.3 Generated Images over Iterations

This subsection provides additional generated images of the trained model across iterations and dataset sizes in a grid format. These images are generated from the vanilla replace paradigm.

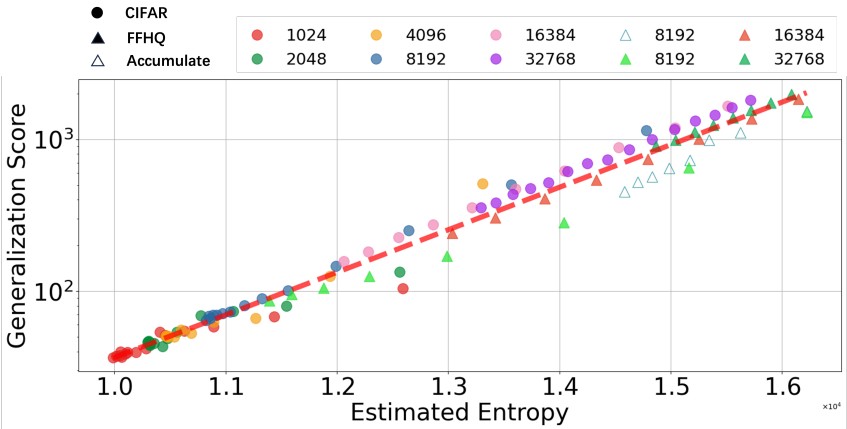

Figure 13: **Scatter plots** of the generalization score and estimated entropy of the training data. Each point denotes one iteration of training in the self-consuming loop. In addition to the results in Figure 4a, we include more results from FFHQ and on the accumulate paradigm. We use different colors to denote the loops, different shapes (circles and triangles) to represent CIFAR and FFHQ, and solid versus hollow markers to distinguish the replace and accumulate paradigms.

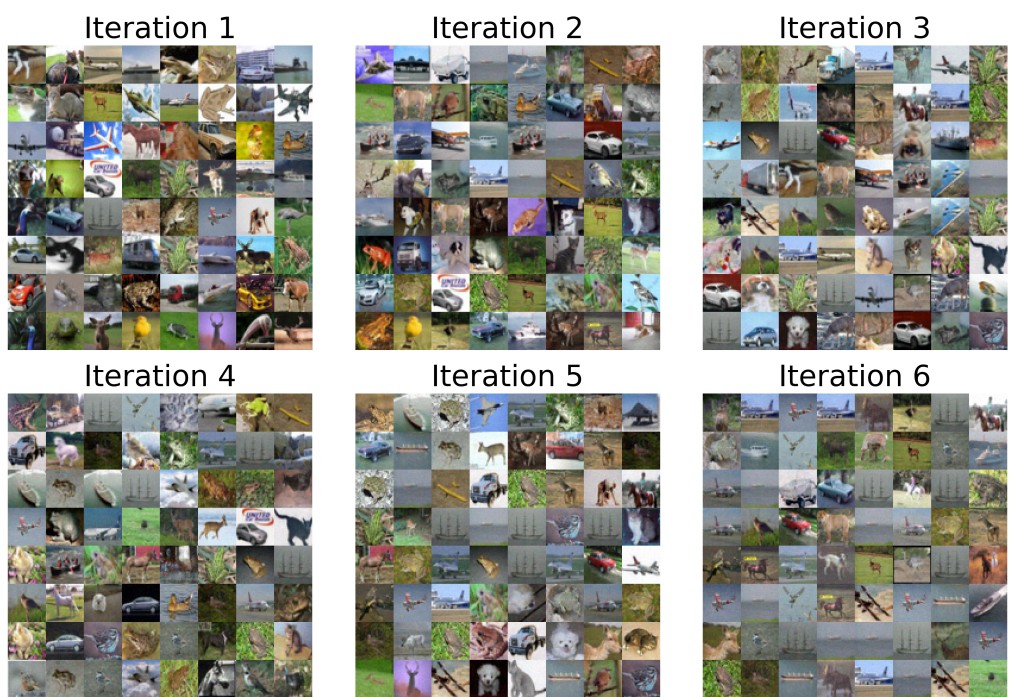

Figure 14: Generated images of models trained on **1024** CIFAR-10 images over iterations.

Figures 14 to 19 present the generated images of the model trained on CIFAR-10 with various dataset sizes across successive iterations. As shown in Figures 14 and 15, the diffusion models tend to memorize small training datasets throughout the recursive process, exactly copying training images during generation. Because the generated samples are nearly identical to the training data, image quality does not noticeably degrade. However, duplicated generations emerge in later iterations, and diversity declines as the model gradually loses coverage over parts of the original images.

On the contrary, on the large dataset of 32,768 images, the models generate novel images in the beginning. As the model collapses, the quality and diversity of the images gradually degrade over iterations.

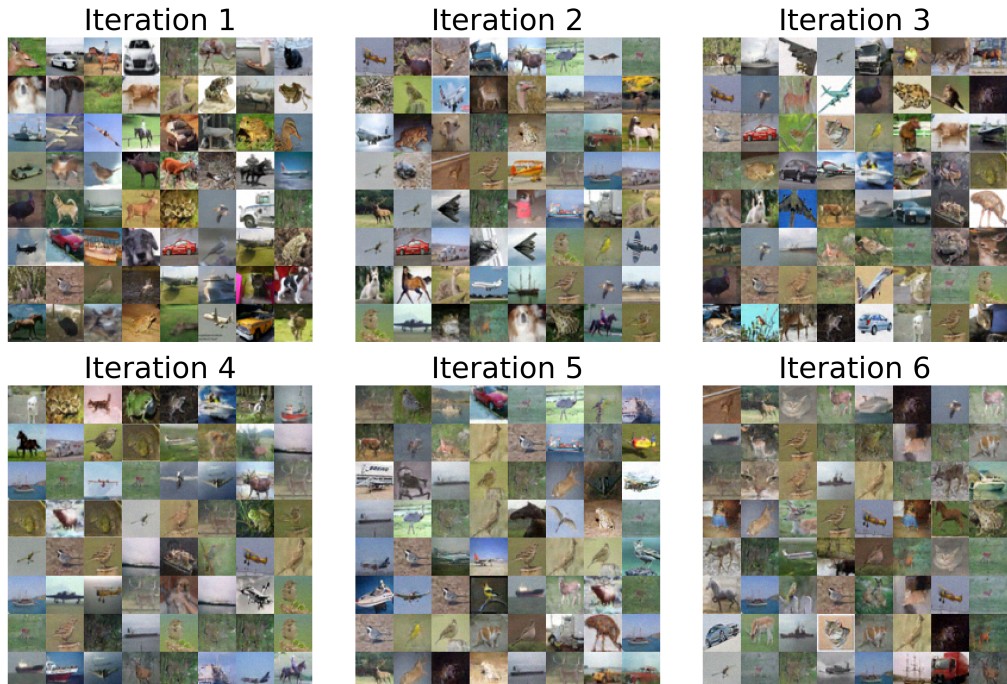

Figure 15: Generated images of models trained on **2048** CIFAR-10 images over iterations.

## D.4 Generated Image Distribution Becomes Spiky

In the main paper, we adopt the KL estimator of Equation (2) to evaluate the entropy given a finite image dataset. With a fixed dataset size, the only dataset-dependent term in Equation (2) is the sum of nearest-neighbor distances $\varepsilon(\boldsymbol{x})$ indicating that samples in $\mathcal{D}$ become increasingly concentrated. This suggests the distribution is becoming spiky. Figure 3 in the main paper also illustrates this trend by projecting high-dimensional images onto the subspace spanned by their top two eigenvectors. The visualization reveals that the generated images form numerous local clusters, while the overall support of the distribution remains relatively stable.

To quantitatively measure the spiky degree of the empirical image distribution, we further adopt the Mean Nearest Neighbor Distance (MNND) [31, 72], which removes the data-independent constants and logarithm in the KL estimator:

$$\texttt{MNND}(\mathcal{D}_t) \triangleq \texttt{Dist}(\mathcal{D}_t, \mathcal{D}_t) = \frac{1}{|\mathcal{D}_t|} \sum_{\boldsymbol{x} \in \mathcal{D}_t} \min_{\boldsymbol{z} \in \mathcal{D}_t \setminus \boldsymbol{x}} d(\boldsymbol{x}, \boldsymbol{z}). \tag{3}$$

A lower MNND suggests a tightly clustered and spiky distribution, while a higher distance suggests dispersion and diversity. The KL estimator can be related to MNND through

$$e^{\frac{\hat{H}_1(\mathcal{D}_t) - B}{d}} \leq \texttt{MNND}(\mathcal{D}_t), \tag{4}$$

where $\hat{H}_1(\mathcal{D}_t)$ represents the estimated entropy of $\mathcal{D}_t$ with $k = 1$ and $B$ is a constant offset given a fixed dataset size.

We want to clarify the nuance between MNND and variance. A spiky distribution does not imply that all data points are concentrated in a single small region—this behavior has already been illustrated by the variance collapse behavior [3–5]. Specifically, Figure 20 shows that the variance of the generated dataset only slightly decreases along the successive iterations and is far from complete collapse. On the contrary, MNND decreases almost exponentially and reaches a small value after 10 iterations. Thus, the results align with the prior claim in [2, 4] that the collapse of variance progresses at such a slow pace that it is rarely a practical concern in real-world applications. In contrast, the collapse in MNND emerges at an early stage, highlighting a critical memorization issue caused by training on synthetic data.

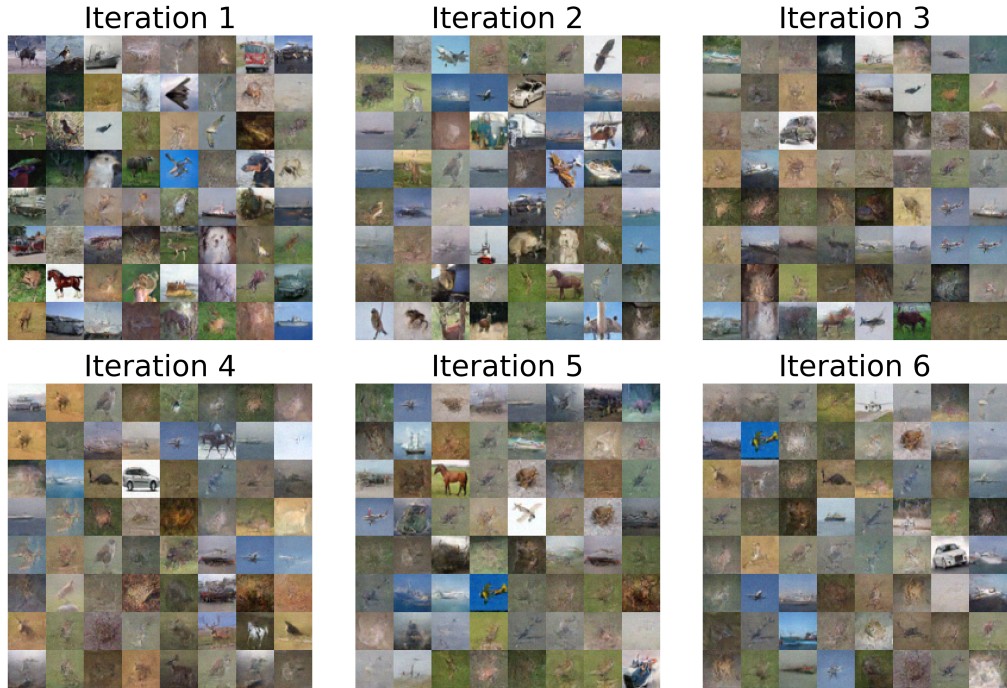

Figure 16: Generated images of models trained on **4096** CIFAR-10 images over iterations.

| Model | A | B | C |
|-------|------|------|------|
| FID | 28.0 | 30.8 | 27.5 |

Table 2: FID comparison of three models.

## D.5 Self-consuming Loop with Fresh New Data

[11] show that incorporating fresh real data can further mitigate model collapse. In this section, we conduct experiments to verify the effectiveness of our methods in this paradigm. The candidate data pool in each iteration is jointly composed of three sources: (1) the original real data, (2) synthetic data generated by the previous model, and (3) fresh real data that was not used in earlier iterations. This experiment is designed to simulate real-world deployment, where generative models are continuously updated with a mixture of prior synthetic data and incoming fresh real data, and the training budget also increases. Concretely, we first train Model A on 32,768 real images, which then generates 10,000 synthetic images. We construct a data poolsimulating an Internet-scale sourceby combining the 32,768 original real images, 10,000 synthetic images, and 10,000 additional fresh real images. From this pool, our entropy-based selection method chooses 40,000 images as the training dataset for Model C. For comparison, Model B is trained on 40,000 randomly selected images from the same pool.

The FID scores of Models A, B, and C are summarized in Table 2. As the results indicate, our entropy-based selection method (Model C) enables the model to outperform the baseline (Model A), and far better than random sampling (Model B). This demonstrates that, in a practical scenario with evolving datasets, our approach not only mitigates model collapse but also delivers tangible performance gains.

## D.6 Additional Metric

Since our Greedy Section method is performed on the feature space extracted by the DINO model, we also use $FD_{DINO}$ [73] as an alternate for FID to measure the quality of the generated images. As

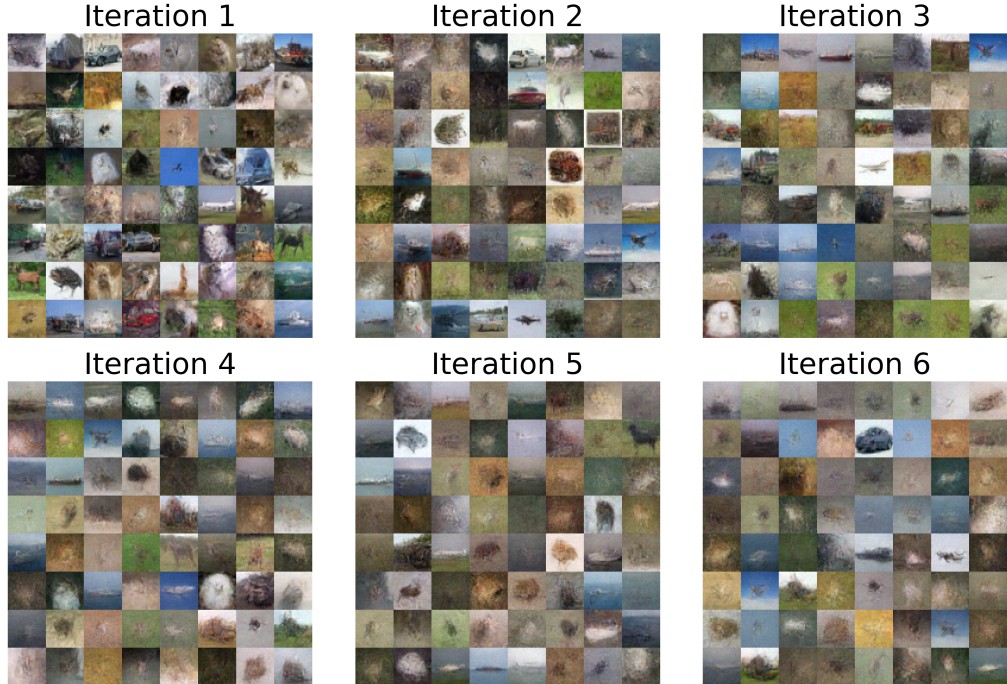

Figure 17: Generated images of models trained on **8192** CIFAR-10 images over iterations.

we show in Figure 21, the Greedy Selection method can also improve the FD$_{\text{DINO}}$ metric compared to the vanilla paradigms.

### D.7 Additional Results for Section 5

We provide visualization of the generated FFHQ images with their nearest training neighbors to show the model's generalizability. Figure 22 shows some images for different training paradigms at 5-th iteration in grid format. With augmentation from the Greedy Selection method, the model generates images that deviate more from the training set compared to the vanilla accumulate paradigm, thereby enhancing its generalization ability.

## E Impact Statement

In this work, we investigate a critical failure mode of diffusion models known as model collapse, which occurs when models are recursively trained on synthetic data and gradually lose their generalization ability and generative diversity. As AI-generated data is unintentionally or deliberately incorporated into the training sets of next-generation models, understanding and mitigating model collapse is essential for ensuring long-term model reliability and performance. Our study identifies the generalization-to-memorization transition, demonstrates the relation between entropy of the training set and the generalizability of the trained model, and proposes practical solutions to mitigate model collapse through entropy-based data selection.

We believe our findings will contribute to the responsible development and deployment of generative models, especially in scenarios where data source may be mixed or partially synthetic. While techniques for analyzing collapse may be misused to intentionally degrade generative models through poisoning attack, our intent is solely to build more robust, transparent, and self-aware AI systems. We encourage researchers in generative AI to use these results to mitigate model collapse and to build reliable models, even when training data contains AI-generated samples—a scenario that may become increasingly common in the future.

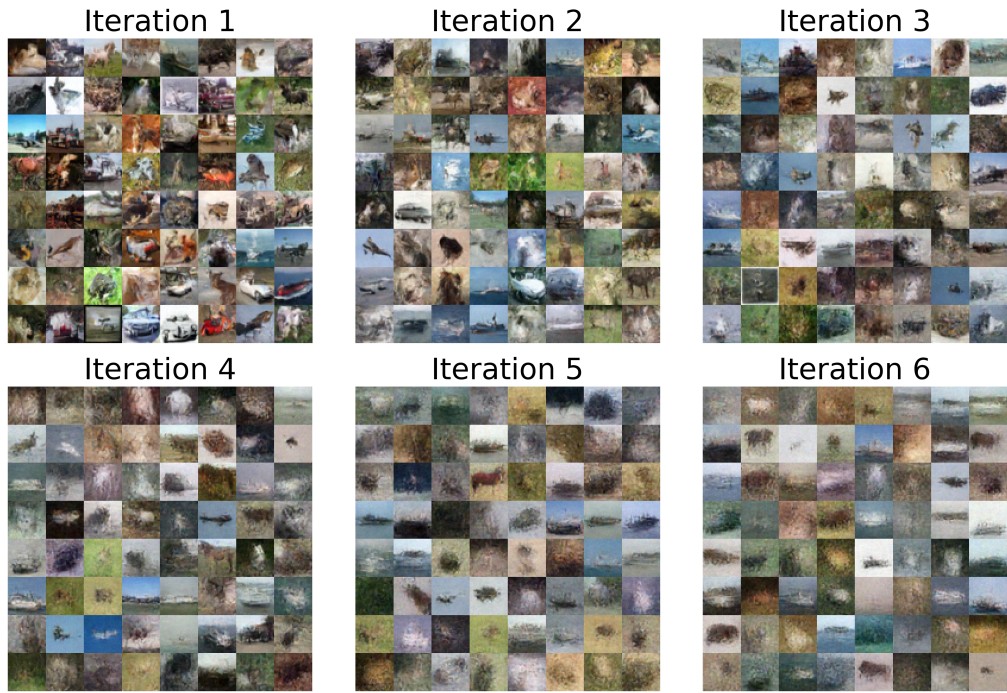

Figure 18: Generated images of models trained on **16384** CIFAR-10 images over iterations.

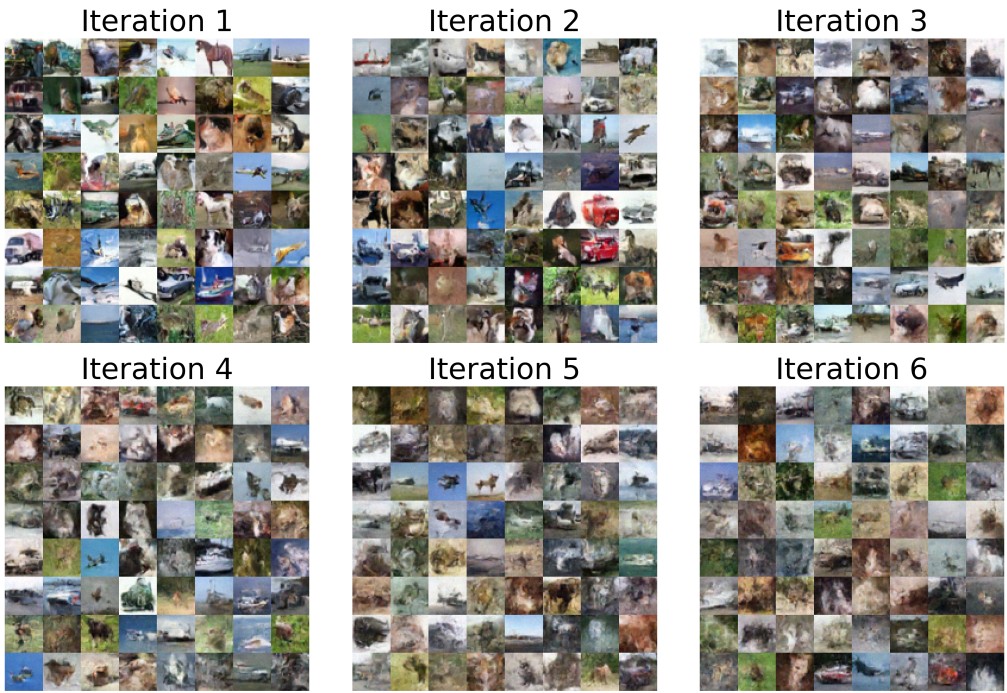

Figure 19: Generated images of models trained on **32768** CIFAR-10 images over iterations.

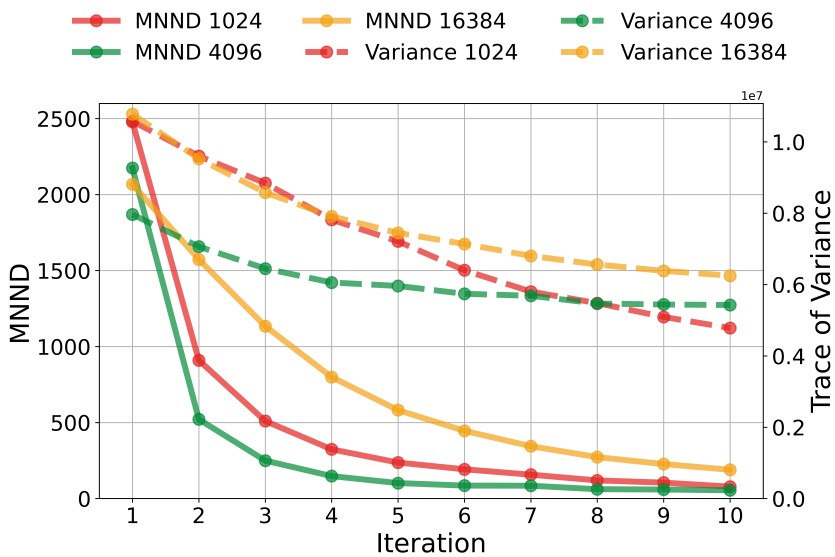

Figure 20: The MNND and the trace of the covariance matrix over iterations.

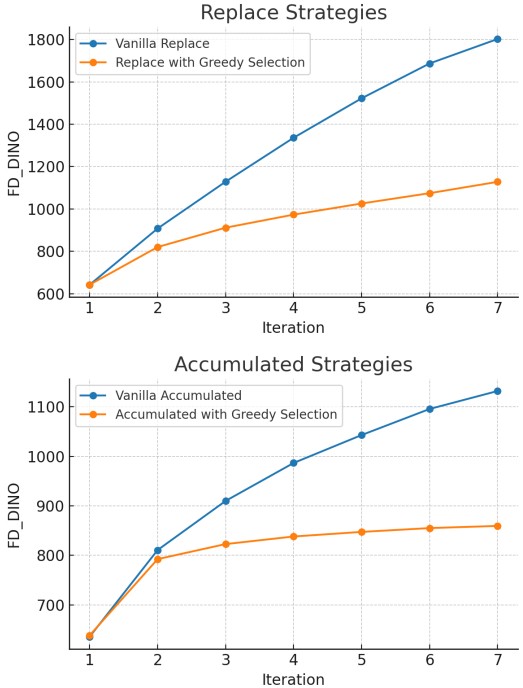

Figure 21: FD$_{\text{DINO}}$ comparison of vanilla paradigms and our methods.

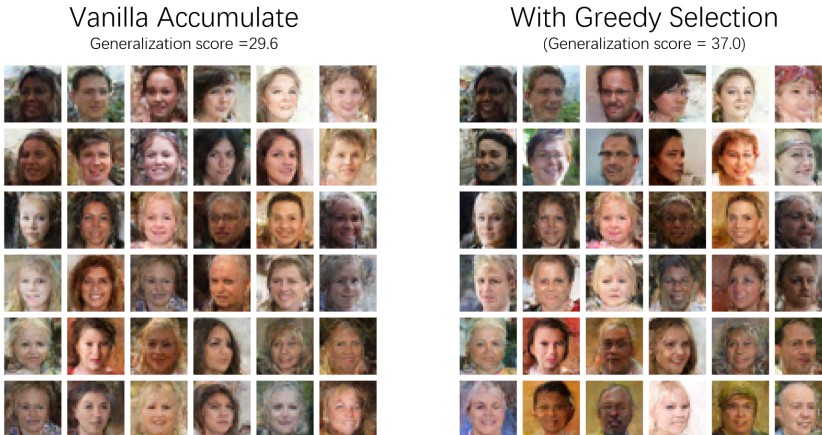

Figure 22: The visualization of the generated images and their nearest neighbors in the training dataset. Each pair of rows corresponds to one group: the top row shows the generated images, and the bottom row shows their nearest training images.

