# OpenReview forum: "A Closer Look at Model Collapse: From a Generalization-to-Memorization Perspective"
_NeurIPS.cc/2025/Conference — NeurIPS 2025 spotlight_

### Official Review · Reviewer_pXga · 2025-06-22

**Clarity:** 3
**Significance:** 2
**Originality:** 3
**Rating:** 5
**Confidence:** 3

**Summary:**

This paper investigates a critical phenomenon in diffusion models: recursively training on self-generated synthetic data leads to a decline in generation quality and, in some cases, memorization of the original training data. Through empirical analysis, the authors demonstrate a strong correlation between the entropy of the training dataset and the model’s generalization capability within this self-consuming training loop. Motivated by this observation, they propose entropy-based data selection strategies that effectively mitigate performance degradation across recursive training iterations.

**Questions:**

## Questions and Comments

### Appendix Section C.1

The results show that the vanilla paradigm using 2N images performs better than using N images, but still worse than the proposed data-selection method applied to N points. This suggests that simply increasing the number of training samples does not necessarily lead to better performance. One possible explanation is that the additional data in the 2N case may contain redundant or low-entropy samples, which could hinder generalization.

It would be helpful to include a more detailed analysis here. For example:
- How do the generalization score and entropy vary across these different training regimes?
- Is it the case that datasets produced by the data-selection method exhibit lower entropy compared to the full 2N image set?
- Would these trends remain consistent across different datasets?

Such analysis could provide deeper insight into the role of data redundancy and entropy in recursive training.

---

### Appendix Section D.1

- Do the generated images in this section come from the vanilla "replace" paradigm?

- When the original training dataset is large, it is observed that after a few recursive training iterations, the generated images—while not exhibiting memorization—become increasingly fuzzy. Do the authors have an intuition for why this occurs? Is this phenomenon consistent across different datasets?

- What visual patterns emerge from applying the proposed data-selection methods over multiple iterations? Do the generated samples still become fuzzy after a few rounds, or is the degradation mitigated?

---

### On the Data Selection Methods

As a disclaimer, I am not an expert in training diffusion models with self-generated data. However, my understanding is that the long-term goal of incorporating synthetic data in recursive training is to improve model performance beyond what is achievable by training solely on the original dataset.

In the current experiments, while the proposed data-selection methods help mitigate performance degradation over iterations, the model's performance in later stages still does not surpass that of the initial training iteration.

To strengthen the paper, it would be highly valuable to include an experiment—possibly unconstrained by fixed dataset size in later iterations but using a fixed neural network architecture—that demonstrates how data-selection strategies can lead to improved performance compared to training only on the original dataset. This would directly support the claim that such methods enhance long-term model generalization.

**Ethical Concerns:**

["NO or VERY MINOR ethics concerns only"]

**Final Justification:**

I have read all the reviewers’ comments and the corresponding replies from the authors (including my own). In my view, all of the points I raised have been satisfactorily addressed. I believe the empirical observations and the proposed method presented in this paper will be of interest to the community. Accordingly, I recommend acceptance.

**Limitations:**

Yes

**Quality:**

2

**Strengths And Weaknesses:**

**Strengths:**

1. The paper is generally well-written and effectively communicates its main takeaways.

2. It presents a compelling empirical observation of a strong correlation between the entropy of the training data and the model's performance during recursive self-training.

3. The inclusion of ablation studies provides meaningful insights into different aspects of the proposed approach.

**Weaknesses:**

1. Some empirical observations would benefit from more detailed analysis and discussion (see in the *Questions* section).

2. The description of the classifier-free diffusion guidance experiment, as well as the explanation of how the proposed data selection methods enhance diversity, lacks depth and could be made more thorough and precise.

---

> ### Author Rebuttal · Authors · 2025-07-31
>
> We appreciate the reviewer for the insightful questions. Below, we provide our response for your comments.
>
> ## Regarding the comments on Appendix C.1:
> **Q1:** ***How do the generalization score and entropy vary across these different training regimes?***
>
> **A1:** In Figures 5 and 7, we have shown the generalization score and entropy of the vanilla "replace" paradigm and the selection-augmented method. The results show that by sampling $2N$ images and then selecting the $N$-size subset, we could improve the entropy of the training dataset, which leads to better generalization. As for training on the $2N$ dataset, we compare the entropy with the data selection method in the next question.
>
> **Q2:** ***Is it the case that datasets produced by the data-selection method exhibit lower entropy compared to the full 2N image set?***
>
> **A2:** No. The $N$ dataset produced by the selection method has **higher** entropy than the full $2N$ dataset. We compute the entropy of the generated $2N$ image dataset and its selected $N$ subset, obtaining values of $746.8$ and $796.4$, respectively. This demonstrates that the selection method yields a subset with higher entropy than the full $2N$ dataset.
>
> By the definition of the KL estimator of entropy:
>
> $$\hat{H}_ \gamma(\mathcal{D})= \psi(|\mathcal{D}|)- \psi(\gamma)+ \log c_ d+ \frac{d}{|\mathcal{D}|}\sum_ {x \in \mathcal{D}} \log \varepsilon_ \gamma(x),$$
>
> the last term represents the **average nearest-neighbor distance**, which is maximized by our selection methods. Although the dataset size decreases, the increase in average distance results in higher entropy.
>
> **Q3:** ***Would these trends remain consistent across different datasets?***
>
> **A3:** We observed these trends on CIFAR-10 and FFHQ in our work. But we think that whether selecting $N$ samples from $2N$ performs better than training on $2N$ also depends on the quality of the synthetic images. If the diffusion model learns pretty well and the generated distribution resembles the underlying image distribution, then more data could diminish the empirical estimation error, and training on the whole $2N$ images leads to better performance. On the other hand, if the quality of the generated samples varies, then our selection method can help us filter out more suitable data.
>
> ## Regarding the comments on Appendix D.1
> **Q4:** ***Do the generated images in this section come from the vanilla "replace" paradigm?***
>
> **A4:** Yes. The images shown in Appendix D.1 are generated from the vanilla "replace" paradigm. We will make this clearer in the revision.
>
> **Q5:** ***When the original training dataset is large, it is observed that after a few recursive training iterations, the generated images—while not exhibiting memorization—become increasingly fuzzy. Do the authors have an intuition for why this occurs? Is this phenomenon consistent across different datasets?***
>
> **A5:** Thank you for highlighting this interesting phenomenon. We have also observed a similar behavior, which was previously reported in [1] (see Figures 14 and 15). Their results show that diffusion models can generate clear images when operating in either the memorization or generalization regime—corresponding to small and large dataset sizes, respectively. However, when the dataset size is around $4096$, lying between the two regimes, the model tends to generate fuzzy images. An intuition is that diffusion models can accurately learn the empirical distribution with small datasets and the underlying distribution with large datasets, but at the boundary between these regimes, the model reaches an unstable solution that neither memorizes nor generalizes well. A rigorous theoretical explanation for this remains an open problem.
>
> Returning to the fuzzy images in Appendix D.1: when the original dataset is large, the initial model generalizes well. However, after several recursive training iterations, the dataset entropy decreases, and the model enters the intermediate region between memorization and generalization, leading to fuzzy generations—consistent with the observations in [1].
>
> [1]: Huijie Zhang et al. "The Emergence of Reproducibility and Consistency in Diffusion Models." Proceedings of the 41st International Conference on Machine Learning, PMLR 235:60558-60590, 2024.
>
> **Q6:** ***What visual patterns emerge from applying the proposed data-selection methods over multiple iterations? Do the generated samples still become fuzzy after a few rounds, or is the degradation mitigated?***
>
> **A6:** The selection method can effectively mitigate the performance degradation compared to the vanilla paradigms, as shown in the FID and generalization improvement. We will also add some generated images for the proposed selection method in the revision to better visualize the improvement. We do not see special visual patterns emerging from our methods.
>
> ## Regarding the improvement of data-selection methods
> **Q7:** ***To strengthen the paper, it would be highly valuable to include an experiment—possibly unconstrained by fixed dataset size in later iterations but using a fixed neural network architecture—that demonstrates how data-selection strategies can lead to improved performance compared to training only on the original dataset. This would directly support the claim that such methods enhance long-term model generalization.***
>
> **A7:** Thank you for the suggestions. We fully agree that the ultimate goal of our method is to **surpass the baseline model** trained on the original dataset. To evaluate our approach in a more realistic scenario—incorporating **fresh data** and an **expanded training data size**—we conducted a new experiment comparing three models:
> 1. **Model A (Baseline)**: The original model trained on $32{,}768$ real images, which then generates $10{,}000$ synthetic images.
> 2. **Model B (Random Sampling)**: A new model trained on a $40{,}000$-image dataset randomly sampled from the pool of original real images, fresh data, and synthetic images.
> 3. **Model C (Our Method)**: A new model trained on a 40,000-image dataset selected from the same pool using our entropy-based method.
>
> The performance of these models is shown below:
>
> | Model        | A | B | C |
> |:------------------:|:----------:|:----------:|:------------:|
> | FID     |   28.0    | 30.8 |   27.5    |
>
> As the results indicate, our entropy-based selection method (Model C) allows the next-generation model to surpass the baseline (Model A), a feat not achieved by random sampling (Model B). This directly shows that in a practical scenario with evolving datasets, our method does not just slow collapse but enables tangible improvement.
>
> The current academic setup for studying model collapse—an iterative loop where no new real data is introduced and the model continuously consumes its own generated data—is valuable for **isolating the collapse phenomenon**, but it does not fully reflect practical usage. Our method is also applicable in **real-world scenarios**, where generative models are continuously updated with a **mixture of previously generated synthetic data and an incoming stream of fresh real data**, yielding improved models.
>
> We believe this addresses the reviewer's concern about practical usefulness.

---

> > ### Comment · Reviewer_pXga · 2025-08-03
> > **Reply to the authors**
> >
> > I would like to thank the authors for providing detailed explanations that address my questions and concerns. In my view, all points I raised have been satisfactorily addressed, and accordingly, I have raised my score.
> >
> > Regarding my question Q2, the comparisons of entropy between N samples selected by the proposed method and full 2N samples aligns with my previous intuition that "the additional data in the 2N case may contain redundant or low-entropy samples, which could hinder generalization." I think it would be benefitial to include a small remark on this point, as it directly reflects the underlying working mechanism of the proposed method.
> >
> > Regarding question Q7, I find the added experiments are both important and valuable, and I recommond that the authors include them in the main paper during revision.

---

> > > ### Author Response · Authors · 2025-08-06
> > >
> > > Thank you for your suggestions. We are glad that your concerns have been addressed. We will incorporate the additional experiments into the revision, and we believe this will greatly enhance the depth and significance of the work.

---

### Official Review · Reviewer_n2Fm · 2025-06-23

**Clarity:** 2
**Significance:** 3
**Originality:** 3
**Rating:** 5
**Confidence:** 3

**Summary:**

If models are trained on the outputs of other models, their performance can decrease in a phenomenon called *model collapse*. This work characterizes model collapse in terms of a transition from generalization to memorization of training data. They find that the entropy of a training dataset is closely related to the generalization score of a model trained on that dataset. Motivated by this finding, they introduce two entropy-based approaches to sub-sampling from real and synthetic datasets, which increase generalization and improve FID scores when models are trained on synthetic and natural data.

**Questions:**

- The authors examine what happens when we train a model on data contaminated with its own generations. In a realistic setting, internet data contains images generated by many different models--how might your analysis change when considering this fact?
- There is a deep question here when looking at Figure 2: why do later iterations of models always collapse to training examples from the *original* train dataset, instead of collapsing to some other random set of synthetic faces? To me this does not seem obvious. Is it that the original "generalized" models actually do have some degree of memorization, which gets amplified in later iterations?
- "projecting high-dimensional images onto the subspace spanned by their top two eigenvectors." -> is this just referring to eigenvectors of the dataset's covariance matrix (i.e., PCA)?

**Ethical Concerns:**

["NO or VERY MINOR ethics concerns only"]

**Final Justification:**

Thank you to the authors for the thorough response. As they have addressed my main questions and concerns on clarity, I will raise my score to 5 (accept).

**Limitations:**

Yes

**Quality:**

3

**Strengths And Weaknesses:**

Strengths:
- Figure 2 provides a strong motivation for why the authors decide to look at entropy of training datasets. Despite the fact that dataset size is constant, they find a consistent decrease in generalization scores, suggesting that the dataset itself contains less and less "information." Their empirical evidence seems to support this interpretation.
- The fact that entropy-aware sampling methods keep generalization high, even in the CIFAR-10 "replace" setting, is quite interesting. This suggests that the culprit for model collapse may be decreased entropy of generative outputs.
- Figure 8 answers a question that I would have asked as a reviewer. The authors interpret it as a good sign that the model over-selects natural images--however, it is also good to see that the algorithm is not *just* filtering out all synthetic images from future runs.
- It is interesting that this approach can also be used to improve CFG results.

Weaknesses:
Many weaknesses of this paper simply have to do with a lack of clarity:
- Figure 2 is confusing due to the word "iteration." Although the authors define it in a footnote on the first page, it would be much clearer to restate this in the Figure 2 caption and in the main body (lines 123-131), and possibly change captions to say "Model-Training Iteration." A quick nod in the caption to why small dataset sizes have much lower generalization scores (even in the first iteration) may help clarify the setup to readers who are skimming.
- Information on models and datasets used should be included earlier in the paper (to understand Section 3 results better).
- Figure 4: Again, more details on the setup in the body and caption would be helpful to the reader. Are different points of the same dataset size re-sampled from the underlying training data? Is there a particular reason why points are connected in 4b but not in 4a?
- Why are DINOv2 features used in Section 4 if they were not needed for Section 3 analyses?
- It would be interesting to see Figure 2 results in the "accumulate" setting, perhaps as an Appendix figure.


The paper could benefit from a little bit more depth, depending on the goal of the authors. If the data sampling method is the central contribution, then how does this approach compare to previous works? If it is the insight that dataset entropy, rather than variance, affects generation, what is the fundamental theoretical difference between these interpretations, and how do your results compare to variance-based sampling? Is this a new observation?

---

> ### Author Rebuttal · Authors · 2025-07-31
>
> **Q1:** ***Figure 2 is confusing due to the word "iteration", although the authors define it in a footnote on the first page.***
>
> **A1:** Thank you for highlighting this. To improve clarity, we will revise the manuscript in two ways. First, we will explicitly define 'iteration' in both the main text and directly in the caption for Figure 2. Second, we will expand the caption to include a brief explanation for why smaller dataset sizes result in lower generalization scores, making the figure easier to interpret.
>
> **Q2:** ***Information on models and datasets used should be included earlier in the paper (to understand Section 3 results better).***
>
> **A2:** Thank you for the important suggestion. In Section 3, we use CIFAR-10 and a UNet-based DDPM model in the experiments. We will provide more details about the models and datasets earlier in Section 3 to improve clarity.
>
> **Q3:** ***Figure 4: Again, more details on the setup in the body and caption would be helpful to the reader. Are different points of the same dataset size re-sampled from the underlying training data? Is there a particular reason why points are connected in 4b but not in 4a?***
>
> **A3:** Thank you for the question regarding Figure 4. This figure plots the generalization score (from Fig. 2) versus the entropy (from Fig. 3) for each of our 54 experiments. We combine the results from Figures 2 and 3 to visualize the relation between entropy and generalization score in Figure 4. Different points of the same dataset size are not just resampled from the underlying data distribution; they denote the results on different iterations of one iterative loop.
>
> The different visualization styles in 4a and 4b were chosen to best highlight the underlying data structure. Specifically, the points in Figure 4a are not connected because they are strongly collinear; connecting them would result in overlapping lines that hide this key trend. We instead use a single dashed line to clearly illustrate this shared linear relationship. Conversely, in Figure 4b, connecting the points is more informative, as it reveals the distinct trends of each group with different data sizes.
>
> **Q4:** ***Why are DINOv2 features used in Section 4 if they were not needed for Section 3 analyses?***
>
> **A4:** We use DINOv2 for the selection method because it yields a slight improvement in FID performance compared to not using DINOv2, likely because FID is also computed in feature space. And it improves computation efficiency since we can select data points in a low-dimensional latent space. We additionally verified that using DINOv2 does not change the observations presented in the analyses in Section 3.
>
> **Q5:** ***It would be interesting to see Figure 2 results in the "accumulate" setting, perhaps as an Appendix figure.***
>
> **A5:** Thanks for the suggestion. We will add the results of the "accumulate" setting as a figure in the Appendix, similar to Figure 2.
>
> **Q6:** ***The paper could benefit from a little bit more depth, depending on the goal of the authors. If the data sampling method is the central contribution, then how does this approach compare to previous works? If it is the insight that dataset entropy, rather than variance, affects generation, what is the fundamental theoretical difference between these interpretations, and how do your results compare to variance-based sampling? Is this a new observation?***
>
> **A6:** We thank the reviewer for this insightful question, which allows us to clarify the central contribution of our work. The primary contribution is the novel insight that **model collapse can be analyzed through a transition from generalization to memorization** and that **data entropy serves as a direct and superior indicator** of this transition compared to variance. This perspective is new, and our proposed data sampling method is the practical application of this core insight.
>
> This addresses the reviewer's questions in the following ways:
>
> 1. **Entropy vs. Variance**: The fundamental difference is that variance measures simple deviation from the mean (which can be skewed by outliers), while entropy measures the 'informativeness' and diversity of the entire data distribution. As our results in Figure 4 already demonstrate, entropy has a stronger linear correlation with generalization, making it a more robust and direct metric for generalization. Based on the reviewer's feedback, we will expand this discussion and incorporate an explicit comparison against a variance-based sampling baseline in the revision.
> 2. **Depth and Future Work**: Our current work provides the essential empirical and conceptual foundation for our claims. Building on this, we will add a discussion of promising future directions for in-depth theoretical studies. We conjecture that the loss of sample diversity is due to the architecture bias and sampling bias of our trained model, and we envision that a more formal theoretical analysis of collapse dynamics can be developed based on learning a mixture of Gaussians. We will add a discussion on these future theoretical directions to the manuscript.
>
> **Q7:** ***In a realistic setting, internet data contains images generated by many different models—how might your analysis change when considering this fact?***
>
> **A7:** Thank you for this insightful comment. We conducted simulation experiments with DDPM and EDM, extending the accumulate-subsample paradigm to a **multi-model** setting. In the first iteration, separate DDPM and EDM models are trained on the real CIFAR-10 dataset. We then generate synthetic images from both models and merge them with the original images to form a sample pool, simulating the Internet pool. In the next iteration, new DDPM and EDM models are trained on a subset sampled from this pool.
>
> The table below reports the FID of these models across iterations in this multi-model scenario. We plan to extend the experiments to more iterations and provide a detailed analysis of the multi-model setting in the revised manuscript
>
> | Iteration        | 1 | 2 |
> |:------------------:|:----------:|:----------:|
> | DDPM | 28.0 | 36.6|
> | EDM | 6.1| 12.6|
>
>
> **Q8:** ***There is a deep question here when looking at Figure 2: why do later iterations of models always collapse to training examples from the original training dataset instead of collapsing to some other random set of synthetic faces?***
>
> **A8:** We would like to clarify that models in later iterations do not necessarily collapse to the original training data points. We analyze this behavior in Appendix D.1, where we visualize generated images across iterations. When the training dataset is small (e.g., $1024$), the model tends to **memorize** the dataset throughout the iterative loop, and later models indeed collapse to the original training images. In contrast, when the dataset is large (e.g., $32768$), early models generate novel images distinct from the original dataset, and later models collapse to **novel points** that are absent in the original data, rather than to the original training images.
>
> **Q9:** ***"projecting high-dimensional images onto the subspace spanned by their top two eigenvectors." -> is this just referring to eigenvectors of the dataset's covariance matrix (i.e., PCA)?***
>
> **A9:** Yes, it is. We use eigen-decomposition to compute the top two eigenvectors of the covariance matrix. This is equivalent to directly performing SVD on the data matrix and getting the top two (right) singular vectors.

---

> > ### Comment · Reviewer_n2Fm · 2025-08-03
> >
> > Thank you to the authors for the thorough response. As they have addressed my main questions and concerns on clarity, I will raise my score to 5 (accept).

---

> > > ### Author Response · Authors · 2025-08-06
> > >
> > > Thank you for your feedback. We are glad that your concerns have been addressed, and we will update the revision based on your suggestions and comments.

---

### Official Review · Reviewer_SEJh · 2025-06-28

**Clarity:** 4
**Significance:** 3
**Originality:** 3
**Rating:** 5
**Confidence:** 4

**Summary:**

This work investigates model collapse in the iterative retraining loop of generative models on their own synthetic data. More precisely, they show that the entropy of the generated data set decreases through the retraining iterations, and that models increasingly memorize their training by generating data in clusters close to the training points. They propose a new method, based on data filtering (and another smoothed out version using a threshold parameter) by iteratively selecting data points based on their distance with others in the training set to ensure higher-entropy in the training set. They empirically show through experiments on image datasets using diffusion models, that this filtering method mitigates model collapse (lower FID divergence) both in the fully-replacement and accumulation scenarios.

**Questions:**

- You chose in your experiments to filter the synthetic dataset size by half (lines 890 to 895). In appendix C.1 you showed that even when comparing to the baseline using twice more data, you method was slowing down model collapse faster. I was wondering if you studied the impact of the filtering fraction (instead of 1/2 perhaps keep only 1/3 of the data or 2/3) and if there was a trade-off appearing where the optimal fraction should be measurable?

- you measured generalization and quality independently using respectively eq1 for the generalization score and FID for the quality. This reminds me of the FLD score in [1] where the authors design a new score that incorporates both notions in the same metric. I think that tracking such metric could be interesting as it could potentially reveal interesting trade-off during training.

[1]: Jiralerspong, Marco, et al. "Feature likelihood divergence: evaluating the generalization of generative models using samples." Advances in Neural Information Processing Systems 36 (2023): 33095-33119.

**Ethical Concerns:**

["NO or VERY MINOR ethics concerns only"]

**Final Justification:**

I believe this work should be accepted.

The contribution is *novel* as the authors identify a new collapse phenomenon related to memorization of generative models which is a very active area of study. They propose an *interesting method* to mitigate it based on selecting high-entropy subsets of synthetic datasets.

The authors *adressed most of my concerns* on related work and some experimental details. They additionally proposed new interesting experiments in the rebuttal.

**Limitations:**

An important limitation in my opinion that the authors didn't discuss is the fact even if the proposed method slows down model collapse, I don't think practicionner have any incentive to use it, since it still decreases the performance of the gerative model compared to the initial one. More precisely, I believe that even if the model collapse litterature is interesting to understand the dangers of training on synthetic data, any method that only mitigates it and does not show improvement on the performance of a model is unlikely to be used in practice as it would be better to just keep the original model and not retrain at all. I think it could be valuable to add a discussion on it, or if the author disagree I would be more than happy to engage in a discussion on this topic.

**Paper Formatting Concerns:**

I didn't notice formatting concern.

**Quality:**

3

**Strengths And Weaknesses:**

**Strength**
- this work identifies a *new collapse phenomenon*, focused on the entropy of the generated data distribution, and different fromprevious studies which were focused on the variance of the generated distribution.
- they propose a new method to mitigate the collapse speed by choosing a subset of the generated data points with high entropy. This can potentially be an impactful method in the training of future generation models.
- the authors performed an interesting ablation study in appendix C, especially C.1 is interesting as it shows that even though this filetring method uses more data than the baseline, this increased sample size does not explain by itself the slow-down in collapse since doubling the sample size in the baseline still degrades faster than with filtering. This is very interesting and was crucial to verify.
- the paper is clearly written, with good background review and clear mathematical definitions of the main concepts (definition 3.1, equation 2)

**Weaknesses**
- some related work is missing, especially regarding the impact of data curation (which is used here in the proposed method) on model collapse. For example [1] shows that curating data based on some reward eventually makes the retraining loop collpase to the highest reward region. It is worth discussing this phenomenon here, although a potentially good thing in your framework is that data points are curated collectively (based on the distance to others) and not just using a pointwise reward function which could prevent this from happening. Your method also makes me think of a litterature on data pruning which shows that pruning synthetic data helps learning. See for example, [2,3,5]. I think it would be interesting to draw some paralell between these works and your pruning method which does not directly prune uninformative samples as they advocate, but prune clusters of samples, which could be potentially related. Finally [5] studies how to prevent model collapse from a theoretical view point which could be compared with your method.
- I am not convinced by the effectiveness of the method proposed here. In particular, the method aims at slowing down model collapse but even with this method, the performance degrade. Hence I don't understand in which scenario practicionners should use this method in the first place? Shouldn't they instead not retrain generative models on synthetic data at all and keep the current trained models that are available? More genrally, I believe that the current litterature on model collapse is interesting but designing methods that only *slow-down* the collapse and do not show any form of imporvement are useless, since they would not be used in any case (practicionner will just keep their current models).


[1]: Ferbach, Damien, et al. "Self-consuming generative models with curated data provably optimize human preferences." arXiv preprint arXiv:2407.09499 (2024).

[2]: Sorscher, Ben, et al. "Beyond neural scaling laws: beating power law scaling via data pruning." Advances in Neural Information Processing Systems 35 (2022): 19523-19536.

[3]: Kolossov, Germain, Andrea Montanari, and Pulkit Tandon. "Towards a statistical theory of data selection under weak supervision." arXiv preprint arXiv:2309.14563 (2023).

[4]: Askari-Hemmat, Reyhane, et al. "Improving the Scaling Laws of Synthetic Data with Deliberate Practice." arXiv preprint arXiv:2502.15588 (2025).

[5]:Fu, Shi, et al. "A theoretical perspective: How to prevent model collapse in self-consuming training loops." arXiv preprint arXiv:2502.18865 (2025).

---

> ### Author Rebuttal · Authors · 2025-07-31
>
> We appreciate the review for the important suggestions and concerns. Below we provide our response for your comments.
>
> **Q1:** ***Some related work is missing.***
>
> **A1:** We appreciate the important related work mentioned in the review and will incorporate a detailed discussion in the revised paper. Specifically, we plan to add the following paragraphs:
>
> "[1, 5] study iterative training processes related to our setting. [1] shows that human preferences can be amplified in an iterative loop: modeling human curation as a reward process, the curated distribution $p_t$ converges to $p^*$ that maximizes the expected reward as $t \to \infty$. In [1], the reward $r(x)$ is a **pointwise** function over individual images, whereas our method can be viewed as a curation strategy that maximizes **dataset-level entropy** at each iteration. Whether the theory in [1] extends to **dataset-wise** rewards $r(S)$, such as dataset entropy, remains an open question. Nonetheless, their intuition aligns with our results that entropy can be iteratively enhanced compared to vanilla methods.
>
> [5] provides the first generalization error bounds for Self-Consuming Training Loops (STLs), showing that mitigating model collapse requires maintaining a non-negligible portion of real data and ensuring recursive stability. In contrast, our work focuses on a specific collapse phenomenon—the generalization-to-memorization transition—and introduces an entropy-based data selection algorithm to mitigate this behavior."
>
>
> "[2, 3, 4] focus on data pruning techniques, but not necessarily in the context of self-consuming loops. While both their approaches and ours demonstrate the benefits of data selection, there are several key differences:
> 1. **Objective and evaluation**: [2, 3, 4] primarily study how pruning improves scaling laws, achieving higher accuracy as dataset size varies. In contrast, our work examines how model performance evolves over an **iterative training loop** with a fixed dataset size, focusing on the generalization-to-memorization transition.
> 2. **Task and criteria**: Prior works focus on **classification tasks**, where sample selection depends on label information. Our method targets **generative modeling**, where selection is based on dataset entropy rather than label-driven criteria. Our objectives also differ: prior work emphasizes classification accuracy, while we address **model collapse from a generalization perspective**, providing a different angle of analysis.
> 3. **Pruning strategy**: Methods in [2, 3, 4] largely rely on **per-sample evaluation**, whereas our approach considers **global dataset structure** and relationships between samples.  While this may lead to increased computational complexity, it opens up new possibilities for designing pruning criteria beyond per-sample evaluation.
> 4. **Entropy definition**: In [4], while the authors use a generative model to sample images, the goal is to improve the classification performance of a classifier. And the entropy used in their method is defined in the prediction space of a classifier, which is different from the entropy of a dataset measured in our work.”
>
> [1]: Ferbach, Damien, et al. "Self-consuming generative models with curated data provably optimize human preferences." arXiv preprint arXiv:2407.09499 (2024).
>
> [2]: Sorscher, Ben, et al. "Beyond neural scaling laws: beating power law scaling via data pruning." Advances in Neural Information Processing Systems 35 (2022): 19523-19536.
>
>
> [3]: Kolossov, Germain, Andrea Montanari, and Pulkit Tandon. "Towards a statistical theory of data selection under weak supervision." arXiv preprint arXiv:2309.14563 (2023).
>
> [4]: Askari-Hemmat, Reyhane, et al. "Improving the Scaling Laws of Synthetic Data with Deliberate Practice." arXiv preprint arXiv:2502.15588 (2025).
>
> [5]:Fu, Shi, et al. "A theoretical perspective: How to prevent model collapse in self-consuming training loops." arXiv preprint arXiv:2502.18865 (2025).
>
> **Q2:** ***I don't understand in which scenario practitioners should use this method in the first place? Shouldn't they instead not retrain generative models on synthetic data at all and keep the current trained models that are available? More generally, I believe that the current literature on model collapse is interesting but designing methods that only slow down the collapse and do not show any form of improvement are useless, since they would not be used in any case (practitioners will just keep their current models).***
>
> **A2:** We thank the reviewer for this insightful question about the practical utility of our method. Our method can be generally applied to improve performance in common, real-world scenarios where generative models are continuously updated with a mixture of previously generated synthetic data and an ongoing stream of **fresh real data**.
>
> We agree with the reviewer's comment that the ultimate goal is to obtain a better model than the previous one. The academic setup in model collapse, where no new real data is introduced, is useful for isolating the model collapse phenomenon but does not reflect how these models would be trained in practice. To demonstrate the effectiveness of our method in a more realistic setting that includes fresh data and an increasing training budget, we conducted a new experiment and compared three models:
> 1. **Model A (Baseline)**: The original model, trained on the initial 32,768 real images, which then generates 10,000 synthetic images.
> 2. **Model B (Random Sampling)**: A new model trained on a 40,000-image dataset randomly sampled from the original real images, new fresh data, and the synthetic data.
> 3. **Model C (Our Method)**: A new model trained on a 40,000-image dataset selected from the same pool using our **entropy-based method**.
>
> The performance of these models is shown below:
> | Model        | A | B | C |
> |:------------------:|:----------:|:----------:|:------------:|
> | FID     |   28.0    | 30.8 |   27.5    |
>
> As the results indicate, our entropy-based selection method (Model C) allows the next-generation model to surpass the baseline (Model A), a feat not achieved by random sampling (Model B). This directly shows that in a practical scenario with evolving datasets, our method does not just slow collapse but enables tangible improvement. We believe this addresses the reviewer's concern about practical usefulness.
>
> **Q3:** ***I was wondering if you studied the impact of the filtering fraction (instead of 1/2 perhaps keep only 1/3 of the data or 2/3) and if there was a trade-off appearing where the optimal fraction should be measurable?***
>
> **A3:** Thank you for this insightful question. The reviewer is correct that this ratio presents a critical trade-off. There are two clear extremes:
> 1. If the ratio is 1, all generated images are used for training. As we show in Appendix C, it still degrades faster than filtering (ratio equals to $1/2$).
> 2. Conversely, if the ratio approaches 0, too few images are selected for training, which detrimentally starves the model of data.
>
> While limited to a single iteration due to time constraints, the following results validate our intuition and demonstrate that an intermediate ratio yields the best performance. To investigate the behavior between these extremes, we will conduct new experiments exploring several intermediate ratios and add a summary of this new analysis to the appendix.
>
> | Ratio        | 0.2 | 0.4 | 0.6 | 0.8 | 1.0 |
> |:------------------:|:----------:|:----------:|:------------:|:------------:|:------------:|
> | FID     |   56.0    | 44.7 |   40.0    |   38.2    | 49.5|
>
> **Q4:** ***This reminds me of the FLD score that incorporates both generalization and quality in the same metric.***
>
> **A4:** Thank you for this important suggestion. We use their implementation from GitHub and measure the FLD score of the generated images over iterations. The following table shows the FLD scores of our method with those of the vanilla paradigm on CIFAR-10. But we are not sure whether FLD would be a suitable metric here to measure the collapse trend here. As the table shows, the FLD of the vanilla accumulate-subsample paradigm even decreases along iterations, suggesting that the generalization and quality are improving, **so FLD does not capture the degradation along iterations**. And the differences across iterations are small, indicating that **FLD is not sensitive to the specific generalization-to-memorization transition** studied in our work.
>
> FLD (Accumulate Paradigm)：
> | Iteration        | 1 | 2 | 3 | 4 | 5 | 6 | 7 |
> |:------------------:|:----------:|:----------:|:------------:|:------------:|:------------:|:------------:|:------------:|
> | Vanilla Accumulate     | 21.2| 22.5| 22.6|19.7 |19.9 | 19.6| 18.4|
> | With Greedy Selection     | 21.2| 22.6| 22.7| 22.8|22.2 |22.4 |22.3 |

---

> > ### Comment · Reviewer_SEJh · 2025-08-03
> >
> > I would like to thank the authors for their thoughtful reply to my questions. I would like to ask for some clarification about some of the answers:
> >
> > **A3**
> > Thank you for these new results. However, I am still quite unsure about whether *iterative retraining*, even with your method, can be useful to a practicionner.  In your experiment, you show improvement over **1 step** of retraining which is quite different from *iterative retraining* where this retraining should be applied a lot of times.
> >
> > Additionally, regarding the results, If I understand correctly, you are claiming that 1 step of your method *improves* of only training on training data, using your filtered synthetic data on top of it? This seems a strong result. Could you clarify the noise level in the experiments (eg noise of the FID over iterations or seeds) and if you managed this result first shot or had to do some hyper-parameter tuning around the number of synthetic images? More generally, I would like to qualitatively know if this result was "easy" to obtain and if you believe it could transfer to other settings? If yes that would be great, but it's also fine if not !
> >
> > **A4**
> > Thank you for reporting the results. Regarding the FLD, I think it is computed using a train and a test set (used to compute distances between generated samples and samples from the training distribution) which in your case should not be CIFAR10. I was thinking they should be set to the previously generated dataset (ie dataset at iteration t-1). Did you use this dataset to compute the FLD? Otherwise, since the collapsed images are not necessarily from the original training set it would "make sense" that the FLD does not increase too much (since distances with CIFAR10 images do not collapse necessarily).
> > On the other hand it is quite surprising that it decreases, even when using CIFAR10 as train set in the FLD.
> >
> > I also had some additional questions:
> >
> > 1) Did you perform retraining from scratch or fine-tuning at each step? It could be nice to add it in the paper.
> >
> > 2) On Figure 5, 6 the accumulate scenario doesn’t seem to converge, even if it slows down collpase. Could you comment on that especially regarding related works showing that accumulating data prevents collapse?

---

> ### Author Response · Authors · 2025-08-06
>
> Thank you for your valuable comments and questions. Below is our response.
> ## Regarding the training experiments
> We would like to clarify the setup of the additional experiment presented in A2 of the rebuttal. **Our training dataset is not simply obtained by adding filtered synthetic data to the original real data.** Instead, it is **selected from a joint pool** composed of three sources: (1) the original real data, (2) synthetic data generated by the previous model, and (3) fresh real data that was not used in earlier iterations. This experiment is designed to **simulate real-world deployment**, where generative models are continuously updated with **a mixture of prior synthetic data and incoming fresh real data**.
> Concretely, we first train **Model A** on 32,768 real images, which then generates 10,000 synthetic images. We construct a data pool—simulating an Internet-scale source—by combining the 32,768 original real images, 10,000 synthetic images, and 10,000 additional fresh real images. From this pool, our entropy-based selection method chooses 40,000 images as the training dataset for **Model C**. For comparison, **Model B** is trained on 40,000 randomly selected images from the same pool.
>
> The FID scores of Models A, B, and C are summarized in the following table.
> | Model        | A | B | C |
> |:------------------:|:----------:|:----------:|:------------:|
> | FID     |   28.0    | 30.8 |   27.5    |
>
> As the results indicate, **our entropy-based selection method (Model C) enables the model to outperform the baseline (Model A), and far better than random sampling (Model B).** This demonstrates that, in a practical scenario with evolving datasets, our approach not only mitigates model collapse but also delivers tangible performance gains. **This experiment was conducted only once, without any hyperparameter tuning;** all hyperparameters match those used in the main paper. For multiple iterations, we are currently conducting the experiment (which is quite expensive) and will post the results as soon as they are available.
>
> ## Regarding the FLD scores
> Thank you for the reminder. The FLD scores reported in the rebuttal were indeed computed by setting the previously generated images as the “training dataset” in the FLD evaluation, rather than the original CIFAR-10 images.
> For completeness, we also tried computing FLD by using the original CIFAR-10 as the “training dataset,” purely out of curiosity, and observed that the FLD scores increase across iterations.
>
> ## Regarding additional questions
> **Q1:** ***Did you perform retraining from scratch or fine-tuning at each step? It could be nice to add it in the paper.***
>
> **A1:** We perform retraining from scratch following the setting of prior model collapse studies. Thanks for your suggestion, and we will clarify it in the revision.
>
> **Q2:** ***On Figure 5, 6 the accumulate scenario doesn’t seem to converge, even if it slows down collpase. Could you comment on that especially regarding related works showing that accumulating data prevents collapse?***
>
> **A2:** We believe one potential reason is that convergence requires more iterations. As shown in Figure 4 of [1], the test errors of LLMs tend to converge only after $30$–$40$ iterations, whereas we conducted fewer than $10$ iterations, since the convergence is not our primary focus.
>
> [1]: Joshua Kazdan, et al. “Collapse or Thrive? Perils and Promises of Synthetic Data in a Self-Generating World.” NeurIPS 2024 Workshops: Mathematics of Modern Machine Learning (M3L) and Attributing Model Behavior at Scale (ATTRIB).

---

> > ### Comment · Reviewer_SEJh · 2025-08-08
> >
> > I thank the authors for their reply to my questions. I am satisfied with their answers and thank them for their additional experiments. In light of this and their answers to other reviewers, I am reinforced in my opinion that this work should be accepted. I will increase my score accordingly.

---

> > > ### Author Response · Authors · 2025-08-09
> > >
> > > We are very glad that our answers addressed your concerns, and we appreciate your important comments. We will accordingly conduct additional comprehensive experiments and incorporate the results into the revision.

---

### Official Review · Reviewer_dMTS · 2025-07-01

**Clarity:** 4
**Significance:** 4
**Originality:** 3
**Rating:** 5
**Confidence:** 4

**Summary:**

This work studies model collapse of successive generative models trained on the previous version’s output. It shows clear evidence of models increasingly replicate training data instead of generating novel content during iterative training on synthetic samples, and proposed an entropy-based data selection technique that can help mitigate the collapse.

**Questions:**

- Given that DINOv2 is used to extract image features for the distance computation, the authors could examine the improved DINOv2 evaluation metrics as presented here rather than FID - https://arxiv.org/abs/2306.04675. With the many known issues with FID, FD_DINOv2 would be a better choice for Figure 6.
- It is interesting that there doesn't seem to be any evidence that supports adding in generated samples helps improve model performance. I understand that in a real use case training on internet data you do not necessarily know what is real or generated and so need automated ways to mitigate collapse, but I wonder if the authors can comment on this. i.e. can the entropy-based data selection result in a better model than naive sampling real images.

**Ethical Concerns:**

["NO or VERY MINOR ethics concerns only"]

**Final Justification:**

My Rating of 5: Accept still holds, as this is a topical ML paper with good experimentation & results. I thank the authors for their additional experiments and clarifications.

**Limitations:**

Yes

**Quality:**

3

**Strengths And Weaknesses:**

- This paper is well written, topical, contains well-designed experiments and reports interesting results.

- The technique introduced to mitigate model collapse has strong supporting evidence of its utility.

---

> ### Author Rebuttal · Authors · 2025-07-31
>
> We thank the reviewer for the detailed and constructive comments.
>
> **Q1:** ***The authors could examine the improved DINOv2 evaluation metrics rather than FID.***
>
> **A1:** We fully agree that incorporating multiple evaluation metrics is important for comprehensively assessing image quality during iterative training. The following tables present the FD_DINOv2 for the generated images over iterations. Our Greedy Selection method yields lower FD_DINOv2 scores in both “replace” and “accumulate” paradigms.
>
> | Algorithm      | Iter 1 | Iter 2 | Iter 3 | Iter 4 | Iter 5 | Iter 6 |  Iter 7 |
> |:-----------:|:----------:|:----------:|:------------:|:----------:|:------------:|:----------:|:------------:|
> | Vanilla Replace     |642.3 | 908.0 | 1128.9 | 1336.3 | 1522.4 | 1686.9 | 1802.3|
> | Replace with Greedy Selection      |642.6 | 820.4 | 911.8 | 973.4 | 1025.9 | 1074.6 | 1128.1|
>
> | Algorithm| Iter 1 | Iter 2 | Iter 3 | Iter 4 | Iter 5 | Iter 6 |  Iter 7 |
> |:------------------:|:----------:|:----------:|:------------:|:----------:|:------------:|:----------:|:------------:|
> | Vanilla Accumulated     |635.5 | 810.7 | 910.0 | 986.7 | 1043.1 | 1095.8 | 1132.1|
> | Accumulated with Greedy Selection  |638.2 | 792.4 | 822.8 | 838.1 | 847.4 | 855.1 | 859.5|
>
> [1]: Jiralerspong, Marco, et al. "Feature likelihood divergence: evaluating the generalization of generative models using samples." Advances in Neural Information Processing Systems 36 (2023): 33095-33119.
>
> **Q2:** ***It is interesting that there doesn't seem to be any evidence that supports adding in generated samples helps improve model performance. I understand that in a real use case training on internet data you do not necessarily know what is real or generated and so need automated ways to mitigate collapse, but I wonder if the authors can comment on this. i.e. can the entropy-based data selection result in a better model than naive sampling real images.***
>
> **A2:** We thank the reviewer for this insightful question, which allows us to clarify the core premise of our work. It raises two key points:
> 1. **Why doesn't adding generated data improve performance in this context?** The reviewer's observation is correct. From a statistical perspective, any real image dataset, $S$, is an i.i.d. sample from the true underlying data distribution, $P$. A generative model, due to practical training and approximation errors, will inevitably learn a slightly shifted distribution, $P′$. Therefore, adding samples from $P′$ to a perfect sample from $P$ will always introduce a distributional shift, leading to a degradation in performance on tasks evaluated against the true distribution, $P$.
> 2. **Can our method outperform naively sampling real images?** In an idealized setting, the answer is no. If a practitioner had access to a massive, perfectly labeled pool of purely real images, exclusively using that data would be the optimal strategy.
> However, our work addresses the more realistic and challenging scenario central to model collapse: a practitioner has a large, **mixed dataset where real and synthetic images are indistinguishable**. The goal is not to outperform a hypothetical 'pure-real' dataset, but to find the best possible **filtering method** to mitigate the negative effects of the synthetic data within this contaminated pool. Our entropy-based selection method is designed for this specific purpose, and as we show, it demonstrably outperforms other sampling strategies in this practical setting.

---

### Note · Authors · 2025-08-12

Dear AC and Reviewers,

We sincerely appreciate your valuable comments and constructive discussions during the rebuttal.

**Review summary**. We are pleased that, following the rebuttal, three reviewers (SEJh, n2Fm, and pXga) raised their scores. Reviewer **dMTS** stated, “The technique introduced to mitigate model collapse has strong supporting evidence of its utility.” Reviewer **SEJh** commented, “I am satisfied with their answers and thank them for their additional experiments. I am reinforced in my opinion that this work should be accepted.” Reviewer **n2Fm** remarked, “They have addressed my main questions and concerns on clarity; I will raise my score to 5 (accept).” Reviewer **pXga** noted, “All points I raised have been satisfactorily addressed.”

Below we **summarize the shared points and our improvements**.

* **Effectiveness and Practical Applicability**: We addressed the core concern of whether our method can outperform a model trained solely on fully real images. With new experiments, we show that in realistic settings with fresh data, our method not only avoids model collapse but also surpasses the original in performance, validating its practical value and robustness.
* **Additional Quality Metrics**: Beyond FID, we report FD_DINOv2 scores, which consistently confirm the quality gains and provide independent validation of our method’s effectiveness.
* **Clarifications**: Following Reviewer n2Fm’s request, we expanded discussions in Sections C.1 and D.1 with added explanations and intuitions and revised figure/table titles to clearly convey experimental settings.
* **Related Work**: Addressing Reviewer SEJh’s point, we analyzed the cited data selection literature and contrasted it with ours across four dimensions: objectives, tasks, pruning strategies, and entropy definitions, highlighting our novel perspective and unique mitigation strategy.

For the **paper revision**, besides addressing the points above, we will further strengthen our paper by:
* Add results of the iterative self-consuming loop with fresh data, showing superiority in realistic applications.
* Incorporate related work suggested by Reviewer SEJh.
* Include results for different filtering ratios for the 2N data pool.
* Add FD_DINOv2 and entropy results.
* Improve clarity in the text and figure/table titles.
* Discuss fuzzy images in Appendix D.1.

Overall, we remain confident in the contributions of this work and thank the AC and reviewers for helping us strengthen it.

---

### Decision · Program_Chairs · 2025-09-17

**Decision:**

Accept (spotlight)

**Comment:**

This paper studies model collapse, wherein models successively trained on synthetic data become progressively worse. The authors identify that the entropy of the generated data decreases as model collapse progresses, and they leverage this insight for proposing mitigation strategies. The reviewers unanimously praised the paper for being topical, relevant, and convincing.

This paper is a clear accept, which I recommend for a spotlight presentation.